# Improving Zero-Shot Adversarial Robustness in Vision-Language Models by Closed-form Alignment of Adversarial Path Simplices

**Junhao Dong** [⋆ 1 2]   **Piotr Koniusz** [⋆ 3 4]   **Yifei Zhang** [1]   **Hao Zhu** [3]   **Weiming Liu** [5]   **Xinghua Qu** [6]   **Yew-Soon Ong** [1 2]

## Abstract

Vision-Language Models (VLMs) such as CLIP excel at zero-shot classification due to large-scale pre-training but are vulnerable to adversarial examples. Adversarial fine-tuning robustifies zero-shot models by aligning prediction scores of individual adversaries with their clean counterparts, which typically overlooks intermediate adversarial samples along the adversarial trajectory crossing the decision boundary. Such intermediate adversaries and their vicinity produce informative representations capturing the decision boundary in detail. They can be improved by sampling adversarial candidates from simplices formed by joining two consecutive vertices on the adversarial trajectory and their clean counterpart. However, sampling simplices for adversaries is very costly. To train robust VLM, we overcome these limitations by Taylor expansion and formulating an upper-bound of alignment loss that depends on the Jacobian/Hessian obtained at clean samples. As regions between clean and intermediate adversarial samples capture a larger decision landscape, we robustify VLM by plausible adversaries from simplices by our closed-form formulation equivalent to infinite uniform sampling of the simplex. We obtain state-of-the-art robustness across 15 datasets and diverse vision-language tasks.

## 1. Introduction

Despite significant advancements driven by Deep Neural Networks (DNNs) in various areas, Szegedy et al. (2014);

---
[1]Nanyang Technological University, Singapore ⋄ [2]Centre for Frontier AI Research, IHPC, A*STAR, Singapore ⋄ [3]Data61♥CSIRO, Australia ⋄ [4]Australian National University ⋄ [5]Zhejiang University, China ⋄ [6]Bytedance, Singapore. [⋆]JD/PK junior/senior leads. PK is also in charge of theory. Correspondence to: Piotr Koniusz <piotr.koniusz@data61.csiro.au>, Yew-Soon Ong <asysong@ntu.edu.sg>.

*Proceedings of the $42^{nd}$ International Conference on Machine Learning*, Vancouver, Canada. PMLR 267, 2025. Copyright 2025 by the author(s).

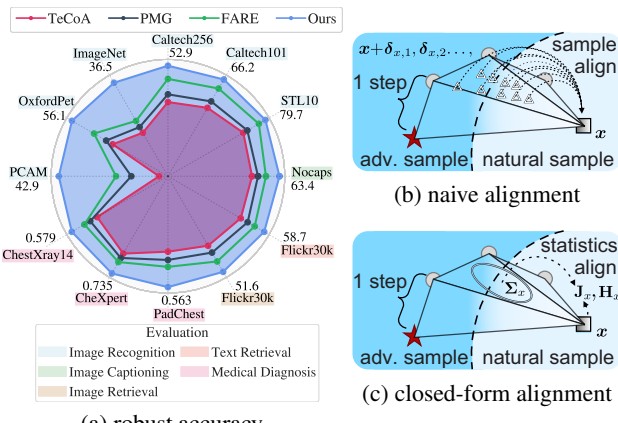

(a) robust accuracy

(b) naive alignment

(c) closed-form alignment

*Figure 1.* Our *AdvSimplex*. Fig. 1a shows the zero-shot robust performance of our method (against adversarial samples) *vs.* adversarial fine-tuning approaches (TeCoA (Mao et al., 2023), PMG (Wang et al., 2024), and FARE (Schlarmann et al., 2024)) across diverse downstream tasks. Fig. 1b shows the principle of sampling simplices formed from vertex **x** and consecutive intermediate adversaries $\mathbf{x}+\boldsymbol{\delta}_{x,i}$ and $\mathbf{x}+\boldsymbol{\delta}_{x,i+1}$ along the adversarial trajectory obtained with several steps of gradient ascent. However, sampling such simplices and aligning them individually with **x** is costly. Fig. 1c shows our model which is much faster as it computes closed-form $\boldsymbol{\Sigma}_x$ used in our alignment formula based on Jacobian and Hessian $\big(J_g(\mathbf{x}), (H_g(\mathbf{x}))_c\big)$.

Goodfellow et al. (2015); Dong et al. (2024a) have shown their vulnerability to adversarial examples, which are low-level perturbations added to legitimate samples to elicit incorrect class predictions. Such adversarial vulnerabilities also affect Vision-Language Models (VLMs) (Zhang et al., 2022a; Zhao et al., 2023a), posing concerns about deploying VLMs in real-life applications (Díaz-Rodríguez et al., 2023).

To counteract malicious adversaries, a growing body of research seeks to strengthen zero-shot adversarial robustness of VLMs through adversarial fine-tuning (Mao et al., 2023; Wang et al., 2024; Dong et al., 2025), with a focus on CLIP-based architectures (Radford et al., 2021). These methods predominantly align single adversarial predictions—derived from feature-level image-text cosine similarity—with either their benign counterparts or the ground-truth labels. However, such an alignment scheme overlooks the broader spectrum of underlying adversaries, especially intermediate

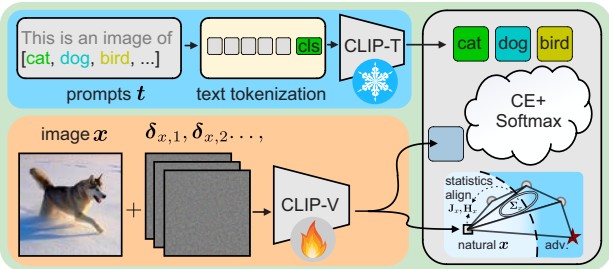

*Figure 2.* The pipeline of our *AdvSimplex*. Class-wise prompts are formulated and passed via frozen CLIP-T (text branch in the blue box). Clean image **x** is passed via CLIP-V (vision branch in the orange box). By the gradient ascent we obtain $\mathbf{x}+\boldsymbol{\delta}_{x,1},\ldots,\mathbf{x}+\boldsymbol{\delta}_{x,m}$ (intermediate) adversarial samples. The alignment process from Fig. 1c is used to form simplices between vertex **x** and consecutive adversarial vertices pairs $(\mathbf{x}+\boldsymbol{\delta}_{x,i},\mathbf{x}+\boldsymbol{\delta}_{x,i+1})$ for $i=1,\ldots,m-1$. Next, an efficient alignment is performed between **x** and all points on simplices to robustify the model.

adversarial samples encountered along *the adversarial trajectory* obtained during iterative adversary generation (*i.e.*, the path that crosses the decision boundary). Although these intermediate adversaries and their variants encode rich information about the class boundaries, limited efforts have been made to incorporate them explicitly during adversarial fine-tuning due to additional computational cost. Consequently, such an oversight exposes VLMs to unforeseen adversaries.

In this work, we exploit the disruptive effect of augmented or diversified adversaries (Wang et al., 2022a; Lu et al., 2023; Li & Spratling, 2023; Dong et al., 2024c; Gao et al., 2024). Li & Spratling (2023) explored data augmentation to improve adversarial diversity. Lu et al. (2023) obtained intermediate adversaries along the adversary generation trajectory to achieve cross-VLM attacks. Gao et al. (2024) extended such a mechanism to a triangular region with the vertices of two consecutive adversarial samples and their clean counterpart. Wang et al. (2022a) exploited such geometric information search for adversaries in the black-box setting. However, the above works focus on conducting attacks with such adversarially diverse sets. Their associated computational cost makes them inapplicable to fine-tuning.

Thus, to effectively exploit the rich structure of decision boundaries and adversarial diversity of samples contained by simplices between a sample vertex and consecutive pairs of intermediate adversary samples, we depart from the traditional point-wise alignment of prediction scores.

Specifically, Gao et al. (2024) sampled adversarial candidates from 2D simplices with vertices $(\mathbf{x},\mathbf{x}+\boldsymbol{\delta}_{x,i},\mathbf{x}+\boldsymbol{\delta}_{x,i+1})$ for adversarial trajectory (with indices $i=1,\ldots,m-1$) obtained by the iterative gradient ascent, and used a small number of such samples for attacks. However, explicit sampling of 2D simplices to robustify VLM is prohibitively costly. Further additional cost is due to passing such samples via

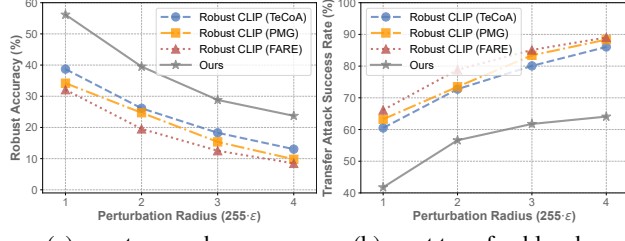

| (a) worst-case adversary | (b) most transferable adv. |

*Figure 3.* Evaluations on ImageNet for our *AdvSimplex vs.* TeCoA (Mao et al., 2023), PMG (Wang et al., 2024), and FARE (Schlarmann et al., 2024). Fig. 3a shows robust accuracy against worst-case adversaries sampled from simplices formed as in Fig. 1a but at test time. The CE loss is used to choose the worst-case adversary. Fig. 3b shows the average transfer attack success rate for the most transferable adversaries. For each model, we choose the most transferable attack from the simplices of remaining models.

the backbone to minimize

$$\min_{\boldsymbol{\theta}} \sum_{\mathbf{x}\in\mathcal{X}} \ell\big(g_{\boldsymbol{\theta}}(\mathbf{x}),y_x\big)+\tfrac{1}{\kappa}\underbrace{\sum_{\boldsymbol{\delta}_x\in\Delta_{\mathcal{X}}}\|g_{\boldsymbol{\theta}}(\mathbf{x}+\boldsymbol{\delta}_x)-g_{\boldsymbol{\theta}}(\mathbf{x})\|_2^2}_{\Omega(\mathbf{x})}.$$

(1)

Here, $\ell(\cdot,\cdot)$ can be cross-entropy (CE) loss, $\kappa=|\Delta_{\mathcal{X}}|$ is the number of adversary candidates sampled from a simplex associated with **x**. Thus, for a dataset with 1M images, and $\kappa=10$ and ascent steps $m=10$, one obtains prohibitive 91M images. Such a naive setting is shown in Fig. 1b.

To train robust VLM, we overcome these limitations by a Taylor expansion of $g_{\boldsymbol{\theta}}(\mathbf{x}+\boldsymbol{\delta}_x)$ around **x** and formulating an upper-bound of approximated alignment loss $\Omega(\cdot)$ that depends on the closed-form $\boldsymbol{\Sigma}_x$ instead of naive computations of costly second-order matrix $\hat{\boldsymbol{\Sigma}}_x=\tfrac{1}{\kappa}\sum_{\boldsymbol{\delta}_x\in\Delta_{\mathcal{X}}}\boldsymbol{\delta}_x\boldsymbol{\delta}_x^{\top}$. We also require easily obtainable Jacobian and Hessian-vector product $\big(J_g(\mathbf{x}),(H_g(\mathbf{x}))_c\,\mathbf{p}\big)$ of $g_{\boldsymbol{\theta}}(\cdot)$ evaluated at a clean sample **x**. The entire pipeline of our model, called *Adversarial Simplex (AdvSimplex)*, is shown in Figure 2. Fig. 1c illustrates our closed-form formulation.

Fig. 1a demonstrates excellent zero-shot robustness of *AdvSimplex* against existing state-of-the-art adversarial fine-tuning approaches on several benchmarks. Moreover, Fig. 3a shows an experiment where a CE loss is used at the test time to select the most disruptive adversarial candidate sample from simplices sharing vertex **x**. Our approach is the most robust model against such adversaries. As TeCoA (Mao et al., 2023), PMG (Wang et al., 2024) and FARE (Schlarmann et al., 2024) perform poorly, this is an indirect proof that such "adversarial simplices" indeed contain adversarial samples. While their strength may differ, they clearly are harmful and thus can be used for robustification. Fig. 3b shows a similar experiment where the most disruptive candidate adversaries were sampled from simplices of all methods other than the tested method to verify cross-model

transferability. The figure shows that the attacks sampled from "adversarial simplices" are both strongly adversarial and highly transferable, highlighting their universality.

In summary, our main contributions are as follows:

1. In contrast to existing adversarial fine-tuning approaches that employ sample-wise prediction alignment of adversaries to clean samples, we employ simplex regions formed from vertex $\mathbf{x}$ and consecutive adversarial pairs $\left\{(\mathbf{x} + \boldsymbol{\delta}_{x,i}, \mathbf{x} + \boldsymbol{\delta}_{x,i+1}) : i = 1, \ldots, m-1\right\}$ from the $m$-steps gradient ascent. While similar schemes were used to create attacks, due to their high computational cost due to sampling, they have not been used in robustification.

2. To alleviate the computational burden of explicit sampling from "adversarial simplices", we derive an upper bound of alignment term $\Omega(\cdot)$ in Eq. (1) by the use of Taylor expansion, and a scalable upper bound that employs closed-form second-order statistic of points contained within simplices, equivalent to infinitely dense uniform sampling strategy. This alleviates the need to pass numerous candidate adversaries through the backbone. Moreover, minimizing our upper bound is shown as minimizing the upper bound of robust risk.

3. We conduct experiments across 15 datasets and diverse scenarios (e.g., Fig. 1), showing that our method outperforms state-of-the-art adversarial fine-tuning approaches.

## 2. Background

Related works are in Section 4. CLIP (Radford et al., 2021) enjoys great performance on zero-shot tasks. Its architecture consists of image and text encoders, parameterized by $\boldsymbol{\theta}$ and $\boldsymbol{\theta}'$. The image encoder $f_{\boldsymbol{\theta}} : \mathcal{X} \to \mathbb{R}^d$ projects input images $\mathbf{x} \in \mathcal{X}$ into a $d$-dimensional feature space. Similarly, the text encoder $f_{\boldsymbol{\theta}'} : \mathcal{T} \to \mathbb{R}^d$ maps input textual descriptions $\mathbf{t} \in \mathcal{T}$ into $d$-dimensional embeddings. By jointly encoding image-text pairs $(\mathbf{x}, \mathbf{t})$, CLIP aligns two modalities by the cosine similarity. The probability of assigning an image $\mathbf{x}$ to a specific category $c \in \{1, \ldots, C\}$ is given as softmax:

$$\left(g_{\boldsymbol{\theta}}(\mathbf{x})\right)_c = \frac{\exp\left(\mathrm{sim}(f_{\boldsymbol{\theta}}(\mathbf{x}), f_{\boldsymbol{\theta}'}(\mathbf{t}_c))\right)}{\sum_{i=1}^{C} \exp\left(\cos(f_{\boldsymbol{\theta}}(\mathbf{x}), f_{\boldsymbol{\theta}'}(\mathbf{t}_i))\right)}, \quad (2)$$

where $\exp(\cdot)$ is the exponential function, and $\mathrm{sim}(\cdot, \cdot)$ represents the cosine similarity. Each text prompt $\mathbf{t}_c =$ "[Context][CLASS$_c$]" (e.g., "This is a photo of a [CLASS$_c$]") is tokenized and embedded by $f_{\boldsymbol{\theta}'}(\cdot)$, serving as the alignment reference. The predicted probabilities of sample $\mathbf{x}$ for all $C$ classes can be represented as $g_{\boldsymbol{\theta}}(\mathbf{x}) = \left[\left(g_{\boldsymbol{\theta}}(\mathbf{x})\right)_1, \ldots, \left(g_{\boldsymbol{\theta}}(\mathbf{x})\right)_C\right]^\top \in [0, 1]^C$.

To enhance zero-shot robustness, adversarial fine-tuning adaptively integrates adversarial samples into the optimization process, aligning their predictions either with their clean

*Table 1.* Summary of key symbols and notations with explanations.

| Symbol | Explanation |
|---|---|
| $\mathbf{x}$ | Clean example. |
| $\boldsymbol{\delta}_{\mathbf{x}}$ | Adversarial perturbation. |
| $g_{\boldsymbol{\theta}}(\cdot)$ | Predictions of the CLIP model. |
| $J_g(\cdot)$ | Jacobian matrix of $g_{\boldsymbol{\theta}}(\cdot)$ (Jacobians stacked for $C$ classes). |
| $\left(H_g(\cdot)\right)_c$ | Hessian matrix of $g_{\boldsymbol{\theta}}(\cdot)$ for class $c$. |
| $\Omega(\cdot)$ | The Euclidean alignment loss. |
| $\bar{\Omega}(\cdot)$ | Upper bound of $\Omega(\cdot)$ with the cross-product term. |
| $\bar{\bar{\Omega}}(\cdot)$ | Upper bound of $\Omega(\cdot)$ without the cross-product term. |

counterparts or one-hot ground-truth labels. Given a set $\mathcal{X}$, the standard adversarial fine-tuning approach (i.e., TeCoA (Mao et al., 2023)) is formulated as a minimax optimization:

$$\min_{\boldsymbol{\theta}} \mathbb{E}_{\mathbf{x} \sim \mathcal{X}} \left[\max_{\|\boldsymbol{\delta}_x\|_\infty \leq \epsilon} \ell\left(g_{\boldsymbol{\theta}}(\mathbf{x} + \boldsymbol{\delta}_x), y_x\right)\right], \quad (3)$$

where $\boldsymbol{\delta}_x$ is an image-level adversarial perturbation constrained within an $\ell_\infty$-norm ball of radius $\epsilon$, and $y_x$ represents the class of $\mathbf{x}$. The inner maximization of the cross-entropy loss generates adversarial examples by perturbing the prediction, while the outer minimization reduces the empirical risk over these adversaries. Following Mao et al. (2023); Wang et al. (2024), adversaries are generated by the iterative Projected Gradient Descent (PGD) (Madry et al., 2018), which updates perturbed input $\hat{\mathbf{x}}^{(i+1)} = \mathbf{x} + \boldsymbol{\delta}_{x,i}$ as:

$$\hat{\mathbf{x}}^{(i+1)} = \Pi_{\mathbb{B}(\mathbf{x}, \epsilon)} \left[\hat{\mathbf{x}}^{(i)} + \alpha \cdot \mathrm{sign}\left(\nabla_{\hat{\mathbf{x}}^{(i)}} \ell\left(g_{\boldsymbol{\theta}}(\hat{\mathbf{x}}^{(i)}), y_x\right)\right)\right], \quad (4)$$

where $\alpha$ denotes the step size, and $\mathrm{sign}(\cdot)$ is the sign function. $\Pi_{\mathbb{B}(\mathbf{x}, \epsilon)}$ ensures the perturbation remains within the $\ell_\infty$-norm ball. Adversarial initialization begins with a random perturbation $\hat{\mathbf{x}}^{(0)} \sim \mathbf{x} + 0.001 \cdot \mathcal{N}(\mathbf{0}, \mathbf{I})$. After $m$ iterations, the final adversarial sample $\hat{\mathbf{x}} = \hat{\mathbf{x}}^{(m)}$ is obtained. The set of the final and intermediate adversarial samples is denoted as $\mathcal{I}_{\mathbf{x}} = \left\{\hat{\mathbf{x}}^{(i)}\right\}_{i=1}^{m}$.

**Problem definition.** Moving beyond traditional robustness evaluations on in-distribution adversarial examples (Croce et al., 2021), we address the more complex *zero-shot* robustness setting (Mao et al., 2023). In this scenario, adversarial examples are generated with unlimited access to data from previously unseen datasets during inference. The goal for defenders, including our approach, is to maintain robustness against these novel security threats, despite having no prior exposure to such data. From a practical defense view, we assume that text prompts, stored in multi-modal systems, remain unchanged during inference, so we do not attack text.

## 3. Proposed Method

Below, we introduce our proposed adversarial fine-tuning approach, enhancing zero-shot adversarial robustness. Our method leverages "adversarial simplices" formed from vertices $(\mathbf{x}, \mathbf{x} + \boldsymbol{\delta}_{x,i}, \mathbf{x} + \boldsymbol{\delta}_{x,i+1})$ for consecutive $i = 1, \ldots, m-1$

given the $m$-steps gradient ascent. Table 1 lists common symbols and their explanations.

## 3.1. Upper-bounding $\Omega(\mathbf{x})$

Below we start with Taylor expansion of $g_{\boldsymbol{\theta}}(\mathbf{x}+\boldsymbol{\delta}_x)$ around $\mathbf{x}$, i.e., $g(\mathbf{x}+\boldsymbol{\delta}_x) = g(\mathbf{x}) + J_g(\mathbf{x})\,\boldsymbol{\delta}_x + \frac{1}{2}\big[\boldsymbol{\delta}_x^T (H_g(\mathbf{x}))_c\,\boldsymbol{\delta}_x\big]_{c=1}^C + \mathcal{O}\big(\|\boldsymbol{\delta}_x\|^3\big)$, where $J_g(\mathbf{x}) \in \mathbb{R}^{C \times wh}$ is the Jacobian matrix evaluated for $g_{\boldsymbol{\theta}}(\cdot)$ at vertex $\mathbf{x}$. We also have $C$ Hessian matrices $\big(H_g(\mathbf{x})\big)_1, \ldots, \big(H_g(\mathbf{x})\big)_C \in \mathbb{R}^{wh \times wh}$ where $wh$ is width $\times$ height of an image. Notice that perturbations $\|\boldsymbol{\delta}_x\|_\infty \leq \epsilon$. Thus, we assume that for a sufficiently small $\epsilon$, the remainder term $\mathcal{O}\big(\|\boldsymbol{\delta}_x\|^3\big)$ of expansion around vertex $\mathbf{x}$ is negligible. Moreover, observe that $\big\|g(\mathbf{x}+\boldsymbol{\delta}_x) - g(\mathbf{x})\big\|_2^2 \approx \big\|J_g(\mathbf{x})\,\boldsymbol{\delta}_x + \frac{1}{2}\big[\boldsymbol{\delta}_x^T (H_g(\mathbf{x}))_c\,\boldsymbol{\delta}_x\big]_{c=1}^C\big\|_2^2$, so for a set $\Delta_{\mathcal{X}}$ with $\kappa$ adversarial perturbations we obtain:

$$\Omega(\mathbf{x}) \approx \frac{1}{\kappa}\sum_{\boldsymbol{\delta}_x \in \Delta_{\mathcal{X}}} \big\|J_g(\mathbf{x})\,\boldsymbol{\delta}_x + \frac{1}{2}\big[\boldsymbol{\delta}_x^T (H_g(\mathbf{x}))_c\,\boldsymbol{\delta}_x\big]_{c=1}^C\big\|_2^2 \quad (5)$$

$$= \frac{1}{\kappa}\sum_{\boldsymbol{\delta}_x \in \Delta_{\mathcal{X}}} \underbrace{\big\|J_g(\mathbf{x})\,\boldsymbol{\delta}_x\big\|_2^2}_{\alpha(\mathbf{x},\boldsymbol{\delta}_x)} + \frac{1}{4}\underbrace{\big\|\big[\boldsymbol{\delta}_x^T (H_g(\mathbf{x}))_c\,\boldsymbol{\delta}_x\big]_{c=1}^C\big\|_2^2}_{\beta(\mathbf{x},\boldsymbol{\delta}_x)}$$

$$+ \underbrace{\big\langle J_g(\mathbf{x})\,\boldsymbol{\delta}_x, \big[\boldsymbol{\delta}_x^T (H_g(\mathbf{x}))_c\,\boldsymbol{\delta}_x\big]_{c=1}^C\big\rangle}_{\gamma(\mathbf{x},\boldsymbol{\delta}_x)} \quad (6)$$

$$\leq \frac{1}{\kappa}\sum_{\boldsymbol{\delta}_x \in \Delta_{\mathcal{X}}} 2\alpha(\mathbf{x},\boldsymbol{\delta}_x) + \frac{1}{2}\beta(\mathbf{x},\boldsymbol{\delta}_x). \quad (7)$$

Eq. (7) is derived from inequality $|x+y|^p \leq 2^{p-1}(|a|^p + |a|^p)$ for $p \geq 1$ due to convexity. See Appendix C.1.

The upper-bound in Eq. (7) does not solve our issue of aggregating over elements of the large set $\Delta_{\mathcal{X}}$. Notice $\alpha(\mathbf{x},\boldsymbol{\delta}_x) = \sum_{c=1}^C \big\langle \boldsymbol{\delta}_x, (J_g(\mathbf{x}))_{c,:}^\top\big\rangle^2 = \sum_{c=1}^C \big\langle \boldsymbol{\delta}_x \boldsymbol{\delta}_x^\top, (J_g(\mathbf{x}))_{c,:}^\top (J_g(\mathbf{x}))_{c,:}\big\rangle$.

As the inner product is linear in its argument, we have:

$$\frac{1}{\kappa}\sum_{\boldsymbol{\delta}_x \in \Delta_{\mathcal{X}}} 2\alpha(\mathbf{x},\boldsymbol{\delta}_x) = 2\sum_{c=1}^C \big\langle \underbrace{\frac{1}{\kappa}\sum_{\boldsymbol{\delta}_x \in \Delta_{\mathcal{X}}}\boldsymbol{\delta}_x \boldsymbol{\delta}_x^\top}_{\hat{\boldsymbol{\Sigma}}_x}, \underbrace{(J_g(\mathbf{x}))_{c,:}^\top (J_g(\mathbf{x}))_{c,:}}_{\mathcal{J}_g(\mathbf{x},c)}\big\rangle \quad (8)$$

We have now arrived at Eq. (8) which depends on $\hat{\boldsymbol{\Sigma}}_x$ (later to be replaced with closed-form $\boldsymbol{\Sigma}_x$) and $J_g(\mathbf{x})$. The case of expanding $\beta(\mathbf{x},\boldsymbol{\delta}_x)$ is more complex and thus we will upper-bound this term. Notice that $\beta(\mathbf{x},\boldsymbol{\delta}_x) = \sum_{c=1}^C \big(\boldsymbol{\delta}_x^T (H_g(\mathbf{x}))_c\,\boldsymbol{\delta}_x\big)^2 = \sum_{c=1}^C \underbrace{\big\langle \boldsymbol{\delta}_x \boldsymbol{\delta}_x^\top, (H_g(\mathbf{x}))_c\big\rangle^2}_{\beta_c(\mathbf{x})}$.

However, $\langle \cdot, \cdot \rangle^2$ is not linear in its arguments, yet we can replace squaring by the Kronecker product $\otimes$ as follows:

$$\beta_c(\mathbf{x},\boldsymbol{\delta}_x) = \underbrace{\big\langle \text{vec}(\boldsymbol{\delta}_x \boldsymbol{\delta}_x^\top) \otimes \text{vec}(\boldsymbol{\delta}_x \boldsymbol{\delta}_x^\top),}_{} \\ \underbrace{\text{vec}((H_g(\mathbf{x}))_c) \otimes \text{vec}((H_g(\mathbf{x}))_c)\big\rangle}_{(\mathcal{H}_g(\mathbf{x}))_c}. \quad (9)$$

Here, $\text{vec}(\cdot)$ simply vectorizes the matrix-shaped input. As each argument of the above inner product contains $(wh)^4$ elements, this is prohibitive. Thus, we use upper-bound:

$$\frac{1}{\kappa}\sum_{\boldsymbol{\delta}_x \in \Delta_{\mathcal{X}}} \frac{1}{2}\beta(\mathbf{x},\boldsymbol{\delta}_x)$$

$$= \frac{1}{2}\sum_{c=1}^C \big\langle \frac{1}{\kappa}\sum_{\boldsymbol{\delta}_x \in \Delta_{\mathcal{X}}}\text{vec}(\boldsymbol{\delta}_x \boldsymbol{\delta}_x^\top) \otimes \text{vec}(\boldsymbol{\delta}_x \boldsymbol{\delta}_x^\top), \mathcal{H}_g(\mathbf{x},c)\big\rangle$$

$$\leq \frac{1}{2}\sum_{c=1}^C \kappa \big\langle \frac{1}{\kappa}\sum_{\boldsymbol{\delta}_x \in \Delta_{\mathcal{X}}}\text{vec}(\boldsymbol{\delta}_x \boldsymbol{\delta}_x^\top) \otimes \frac{1}{\kappa}\sum_{\boldsymbol{\delta}_x \in \Delta_{\mathcal{X}}}\text{vec}(\boldsymbol{\delta}_x \boldsymbol{\delta}_x^\top), \mathcal{H}_g(\mathbf{x},c)\big\rangle$$

$$= \frac{1}{2}\kappa\sum_{c=1}^C \big\langle \underbrace{\frac{1}{\kappa}\sum_{\boldsymbol{\delta}_x \in \Delta_{\mathcal{X}}}\boldsymbol{\delta}_x \boldsymbol{\delta}_x^\top}_{\hat{\boldsymbol{\Sigma}}_x}, (H_g(\mathbf{x}))_c\big\rangle^2. \quad (10)$$

Putting together Eq. (8) & (10), under negligible $\mathcal{O}\big(\|\boldsymbol{\delta}_x\|^3\big)$, we readily obtain $\Omega(\mathbf{x}) \leq \bar{\Omega}(\mathbf{x}) \leq \bar{\bar{\Omega}}(\mathbf{x})$:

$$\bar{\Omega}(\mathbf{x};\hat{\boldsymbol{\Sigma}}_x) = \sum_{c=1}^C \big\langle \hat{\boldsymbol{\Sigma}}_x, \mathcal{J}_g(\mathbf{x},c)\big\rangle + \frac{1}{4}\kappa\big\langle \hat{\boldsymbol{\Sigma}}_x, (H_g(\mathbf{x})_c\big\rangle^2 (+\gamma \text{ terms}),$$

$$\bar{\bar{\Omega}}(\mathbf{x};\hat{\boldsymbol{\Sigma}}_x) = \sum_{c=1}^C 2\big\langle \hat{\boldsymbol{\Sigma}}_x, \mathcal{J}_g(\mathbf{x},c)\big\rangle + \frac{1}{2}\kappa\big\langle \hat{\boldsymbol{\Sigma}}_x, (H_g(\mathbf{x})_c\big\rangle^2. \quad (11)$$

Having developed two upper bounds of $\Omega(\mathbf{x})$, we now define a closed-form expression for $\boldsymbol{\Sigma}_x$.

## 3.2. The $\infty$-dense sampling of simplex (closed-form $\boldsymbol{\Sigma}_x$)

To avoid aggregating empirical $\hat{\boldsymbol{\Sigma}}_x$ over $\kappa$ elements of set $\Delta_{\mathcal{X}}$, we propose the following theorem.

**Theorem 3.1.** *The closed-form expression for* $\boldsymbol{\Sigma}_x = \mathbb{E}\big[\mathbf{p}\mathbf{p}^T\big]$ *over all* $\mathbf{p}$ *in simplex with vertices* $(\mathbf{x},\mathbf{y},\mathbf{z})$ *is*

$$\boldsymbol{\Sigma}_x = \frac{1}{12}\big(\mathbf{x}+\mathbf{y}+\mathbf{z}\big)\big(\mathbf{x}+\mathbf{y}+\mathbf{z}\big)^T + \frac{1}{12}\big(\mathbf{x}\mathbf{x}^T + \mathbf{y}\mathbf{y}^T + \mathbf{z}\mathbf{z}^T\big). \quad (12)$$

*Moreover, let* $Q$ *be the number of vertices* $(\mathbf{z}_1, \ldots, \mathbf{z}_Q)$ *of a simplex. The closed-form expression for* $\boldsymbol{\Sigma}_x = \mathbb{E}\big[\mathbf{p}\mathbf{p}^T\big]$ *for higher-order simplices,* e.g., *tetrahedron (*$Q=4$ *vertices) or pentachoron (*$Q=5$ *vertices) is given as:*

$$\boldsymbol{\Sigma}_x = \frac{1}{Q(Q+1)}\bigg[\sum_{i=1}^Q \mathbf{z}_i\mathbf{z}_i^\top + \Big(\sum_{i=1}^Q \mathbf{z}_i\Big)\Big(\sum_{i=1}^Q \mathbf{z}_i\Big)^\top\bigg]. \quad (13)$$

*Proof.* Parameterize a point $\mathbf{p} \in \mathbb{R}^{wh}$ over vertices $(\mathbf{x},\mathbf{y},\mathbf{z})$ as $\mathbf{p} = \alpha\mathbf{x} + \beta\mathbf{y} + \gamma\mathbf{z}, \alpha, \beta, \gamma \geq 0, \alpha+\beta+\gamma = 1$. Expand $\mathbf{p}\mathbf{p}^T = (\alpha\mathbf{x}+\beta\mathbf{y}+\gamma\mathbf{z})(\alpha\mathbf{x}+\beta\mathbf{y}+\gamma\mathbf{z})^T$ and note that $\mathbb{E}\big[\mathbf{p}\mathbf{p}^T\big] = \mathbb{E}\big[\alpha^2\big]\mathbf{x}\mathbf{x}^T + \mathbb{E}\big[\beta^2\big]\mathbf{y}\mathbf{y}^T + \mathbb{E}\big[\gamma^2\big]\mathbf{z}\mathbf{z}^T + \mathbb{E}\big[\alpha\beta\big](\mathbf{x}\mathbf{y}^T + \mathbf{y}\mathbf{x}^T) + \mathbb{E}\big[\alpha\gamma\big](\mathbf{x}\mathbf{z}^T + \mathbf{z}\mathbf{x}^T) + \mathbb{E}\big[\beta\gamma\big](\mathbf{y}\mathbf{z}^T + \mathbf{z}\mathbf{y}^T)$. For a uniform distribution on the simplex $\{\alpha, \beta, \gamma \geq 0, \ \alpha+\beta+\gamma \leq 1\}$, we have $\mathbb{E}[\alpha] = \mathbb{E}[\beta] = \mathbb{E}[\gamma] = \frac{1}{3}, \mathbb{E}[\alpha^2] = \mathbb{E}[\beta^2] = \mathbb{E}[\gamma^2] = \frac{1}{6}, \mathbb{E}[\alpha\beta] = \mathbb{E}[\alpha\gamma] = \mathbb{E}[\beta\gamma] = \frac{1}{12}$. Substitute expectations into the expansion to conclude the proof for $Q = 3$.

For higher-order simplices (*i.e.*, $Q > 3$), one can parameterize $\mathbf{p} = \sum_{i=1}^{Q} \alpha_i \mathbf{z}_i$, $\alpha_i \geq 0$, $\sum_{i=1}^{Q} \alpha_i = 1$, and obtain: $\mathbb{E}[\alpha_i] = \frac{1}{Q}$, $\mathbb{E}[\alpha_i^2] = \frac{2}{Q(Q+1)}$, and $\mathbb{E}[\alpha_i \alpha_j] = \frac{1}{Q(Q+1)}$, $i \neq j$ for the underlying Dirichlet distribution. Then one expands $\mathbb{E}[\mathbf{pp}^T]$ for $\mathbf{pp}^T = \sum_{i,j=1}^{Q} \alpha_i \alpha_j \mathbf{z}_i \mathbf{z}_j^\top$. $\square$

Eq. (12) requires mere aggregation over four outer products of vectors. As long as $\kappa > 4$, using the closed form is more efficient and equivalent to evaluating for $\kappa = \infty$ set $\Delta_{\mathcal{X}}$.

For efficiency, we use the Hessian-vector product (HVP) in $\bar{\Omega}$ and $\bar{\bar{\Omega}}$, *i.e.*, $(\mathbf{H}_g \cdot \mathbf{p})$ as follows:

$$\langle \mathbb{E}[\mathbf{pp}^T], \mathbf{H}_g \rangle = \mathbb{E}[\mathbf{p}^\top (\mathbf{H}_g \cdot \mathbf{p})] \tag{14}$$

$$= \frac{1}{Q(Q+1)} \left[ \sum_{i=1}^{Q} \mathbf{z}_i^\top (\mathbf{H}_g \cdot \mathbf{z}_i) + \left( \sum_{i=1}^{Q} \mathbf{z}_i \right)^\top \left( \mathbf{H}_g \cdot \sum_{i=1}^{Q} \mathbf{z}_i \right) \right].$$

Thus, we set $\langle \hat{\mathbf{\Sigma}}_x, (H_g(\mathbf{x}))_c \rangle^2 = (\mathbb{E}[\mathbf{p}^\top ((H_g(\mathbf{x}))_c \cdot \mathbf{p})])^2$. For a simplex with $Q = 3$ vertices (one vertex equals $\mathbf{0}$), this requires only three Hessian-vector products. One HPV evaluation costs the same as 2–4 Jacobian evaluations.

### 3.3. Our loss function

Following Fig. 1c, we have to align a set of $m-1$ simplices with the Jacobian and Hessian statistics. Thus, our loss takes the form below:

$$\min_{\boldsymbol{\theta}} \sum_{\mathbf{x} \in \mathcal{X}} \ell(g_{\boldsymbol{\theta}}(\mathbf{x}), y_x) + \lambda \sum_{i=1}^{m-1} \omega_i(\mathbf{x}) \bar{\bar{\Omega}}(\mathbf{x}; \mathbf{\Sigma}_{x,i}), \tag{15}$$

and $\mathbf{\Sigma}_{x,i}$ is evaluated on vertices $(\mathbf{0}, \boldsymbol{\delta}_{x,i}, \boldsymbol{\delta}_{x,i+1})$ by Eq. (12), $\lambda \geq 0$ controls the impact of minimizing the upper bound. Additionally, we can reweight the impact of each simplex by a simple perturbance impact measure of intermediate adversarial vertex $\hat{\mathbf{x}}^{(i)} = \mathbf{x} + \boldsymbol{\delta}_{x,i}$:

$$\omega_i(\mathbf{x}) = \frac{1}{\tau} \left| (g_{\boldsymbol{\theta}}(\mathbf{x}))_{y_x} - (g_{\boldsymbol{\theta}}(\hat{\mathbf{x}}^{(i)}))_{y_x} \right|, \tag{16}$$

where $\tau = \max_{j \in \mathcal{B}} \left| (g_{\boldsymbol{\theta}}(\mathbf{x}_j))_{y_{x_j}} - (g_{\boldsymbol{\theta}}(\hat{\mathbf{x}}_j^{(i)}))_{y_{x_j}} \right|$ simply measures the biggest perturbation for a batch $\mathcal{B}$ of samples $\mathbf{x}$.

### 3.4. Bounding the robust risk

Generally, there exists an inevitable trade-off between natural performance and adversarial robustness (Wang et al., 2024). Below, we study the implications of our design for robust risk (Zhang et al., 2019). Generally, the following three risks are known in adversarial learning:

$$R_{\text{nat}}(g) := \mathbb{E}_{\mathbf{x} \sim \mathcal{X}}[\ell(g(\mathbf{x}), y_x)], \tag{17}$$

$$R_{\text{rob}}(g) := \mathbb{E}_{\mathbf{x} \sim \mathcal{X}} \left[ \max_{\|\boldsymbol{\delta}_x\| \leq \epsilon} \ell(g(\mathbf{x} + \boldsymbol{\delta}_x), y) \right], \tag{18}$$

$$R_{\text{boundary}}(g; \epsilon) := \mathbb{P}_{\mathbf{x} \sim \mathcal{X}} \left[ \exists \boldsymbol{\delta}_x : \|\boldsymbol{\delta}_x\| \leq \epsilon, g(\mathbf{x}) \neq g(\mathbf{x} + \boldsymbol{\delta}_x) \right], \tag{19}$$

where for say the 0-1 loss $\ell(\cdot, \cdot)$, $R_{\text{nat}}(g)$ (the natural risk) is the probability of misclassification on clean data, $R_{\text{rob}}(g)$ (the robust risk) is the probability that the strongest perturbation in $\|\boldsymbol{\delta}_x\| \leq \epsilon$ will cause misclassification. $R_{\text{boundary}}(g; \epsilon)$ quantifies the fraction of points "within $\epsilon$" of the classifier's decision boundary, *i.e.*, the set of points that can be flipped by a perturbation of size $\epsilon$. The prior knowledge also states that the following bound holds $R_{\text{rob}}(g) \leq R_{\text{nat}}(g) + R_{\text{boundary}}(g; \epsilon)$.

For a single adversarial simplex, our boundary risk becomes:

$$R_{\text{boundary}}(g; \Delta_{\mathcal{X}}) := \mathbb{P}_{\mathbf{x} \sim \mathcal{X}} \left[ \exists \boldsymbol{\delta}_x \in \Delta_{\mathcal{X}}, g(\mathbf{x}) \neq g(\mathbf{x} + \boldsymbol{\delta}_x) \right], \tag{20}$$

We can upper-bound this risk by our defined boundary counter risk, which not only captures decision flips on $\mathbf{x}$ but also counts in how many different ways $\mathbf{x}$ can be perturbed to cause a decision flip:

$$R_{\text{counter}}(g; \Delta_{\mathcal{X}}) := \mathbb{E}_{\mathbf{x} \sim \mathcal{X}} \left[ \sum_{\boldsymbol{\delta}_x \in \Delta_{\mathcal{X}}} g(\mathbf{x}) \neq g(\mathbf{x} + \boldsymbol{\delta}_x) \right]. \tag{21}$$

It is clear that $R_{\text{counter}}(g; \mathcal{I}_x) \leq R_{\text{counter}}(g; \Delta_{\mathcal{X}})$ where $\mathcal{I}_x = \{\hat{\mathbf{x}}^{(i)}\}_{i=1}^{m}$ is the set of the final and intermediate adversarial samples of $\mathbf{x}$. As the following holds

$$R_{\text{rob}}(g) \leq R_{\text{nat}}(g) + R_{\text{boundary}}(g; \Delta_{\mathcal{X}}) \tag{22}$$

$$\leq R_{\text{nat}}(g) + R_{\text{counter}}(g; \mathcal{I}_x) \tag{23}$$

$$\leq R_{\text{nat}}(g) + R_{\text{counter}}(g; \Delta_{\mathcal{X}}), \tag{24}$$

we are optimizing the upper bound of the robust risk, taking into account counts of successful perturbations per $\mathbf{x}$.

## 4. Related Works

**Prediction alignment.** As a means of enforcing consistency between model outputs, prediction alignment is widely adopted in machine learning. Originally explored in knowledge distillation (Hinton et al., 2015), where the student network is trained to align its soft predictions with those of a teacher, this concept has since been extended to various domains, including semi-supervised learning (Sohn et al., 2020), unsupervised learning (He et al., 2020), and domain adaptation (Tzeng et al., 2017). Noteworthy are also feature alignment-based domain adaptation (Tas & Koniusz, 2018), few-shot detection and segmentation (Zhang et al., 2022b; Kang et al., 2023; Lu et al., 2024), contrastive learning (Zhang et al., 2025), and misalignment-based anomaly detection (Ding et al., 2025).

In the context of VLMs, prediction alignment plays a critical role in promoting modality consistency and label-space agreement, where recent works leverage alignment losses to improve cross-modal generalization (Jia et al., 2021; Yang

*Table 2.* Zero-shot **clean** accuracy (%). Adversarial fine-tuning is conducted on ImageNet with evaluations across 15 datasets.

| Method | ImageNet | STL10 | CIFAR-10 | CIFAR-100 | SUN397 | Stanf.Cars | Food101 | OxfordPet | Flower102 | DTD | EuroSAT | FGVC | PCAM | Caltech101 | Caltech256 | Average |
|---|---|---|---|---|---|---|---|---|---|---|---|---|---|---|---|---|
| Standard CLIP | 59.13 | 97.17 | 88.55 | 62.29 | 57.68 | 52.07 | 83.84 | 87.35 | 65.60 | 40.05 | 38.31 | 20.13 | 52.26 | 87.08 | 82.01 | 64.90 |
| TeCoA | 54.43 | 91.10 | 72.77 | 41.31 | 44.71 | 22.06 | 39.27 | 75.06 | 38.36 | 29.46 | 22.91 | 10.55 | 42.37 | 77.11 | 70.91 | 48.83 |
| PMG-FT | 51.33 | 90.70 | 73.05 | 42.04 | 44.40 | 27.72 | 42.61 | 75.39 | 39.37 | 29.02 | 20.32 | 11.45 | 47.22 | 80.08 | 71.01 | 49.71 |
| FARE | 50.94 | 93.90 | 81.98 | 56.25 | 49.94 | 42.73 | 63.30 | 81.59 | 53.29 | 34.27 | 21.53 | 14.66 | 45.04 | 85.72 | 75.03 | 56.68 |
| *AdvSimplex* | **61.28** | **96.30** | **87.14** | **59.05** | **53.61** | **43.86** | **69.83** | **84.71** | **56.28** | **37.08** | **23.27** | **16.52** | **48.57** | **87.33** | **78.65** | **60.23** |

*Table 3.* Zero-shot **robust** accuracy (%). Adversarial samples are created using the PGD-20 attack method with the image-level **perturbation radius $\epsilon = 2/255$**. Adversarial fine-tuning is conducted on ImageNet with evaluations across 15 datasets.

| Method | ImageNet | STL10 | CIFAR-10 | CIFAR-100 | SUN397 | Stanf.Cars | Food101 | OxfordPet | Flower102 | DTD | EuroSAT | FGVC | PCAM | Caltech101 | Caltech256 | Average |
|---|---|---|---|---|---|---|---|---|---|---|---|---|---|---|---|---|
| Standard CLIP | 0.54 | 21.38 | 2.21 | 0.73 | 0.35 | 0.20 | 6.11 | 2.99 | 0.54 | 0.22 | 0.03 | 0.00 | 0.00 | 13.60 | 8.81 | 3.85 |
| TeCoA | 27.03 | 71.60 | 44.34 | 23.13 | 18.67 | 5.13 | 13.29 | 42.33 | 16.29 | 17.52 | 12.28 | 2.52 | 13.47 | 56.82 | 45.52 | 27.33 |
| PMG-FT | 26.20 | 71.91 | 45.72 | 23.41 | 18.87 | 6.89 | 14.62 | 43.14 | 16.72 | 18.02 | 12.59 | 2.73 | 21.39 | 58.25 | 46.21 | 28.44 |
| FARE | 24.57 | 77.28 | 52.91 | 30.37 | 17.85 | 9.66 | 18.20 | 46.26 | 18.46 | 19.53 | 10.24 | 2.82 | 23.36 | 62.95 | 49.59 | 30.94 |
| *AdvSimplex* | **36.48** | **79.67** | **57.24** | **32.60** | **20.33** | **11.46** | **20.67** | **56.11** | **19.75** | **20.76** | **13.19** | **4.94** | **42.93** | **66.23** | **52.88** | **35.68** |

et al., 2022; Dong et al., 2025). Departing from conventional point-wise alignment, our method introduces robust alignment over adversarial path simplices (sets of adversarial points), promoting stronger robust generalization.

**Uni-modal adversarial robustness.** The growing use of DNNs in both vision and language tasks (Khan et al., 2022; Wang et al., 2022b; Hu et al., 2024; Zhao & Zhang, 2024) has heightened awareness of their vulnerability to adversarial inputs, stimulating research on defense mechanisms (Bai et al., 2021; Aldahdooh et al., 2022; Xie & Yuille, 2020), among which adversarial training is a very effective paradigm. By iteratively optimizing models against worst-case perturbations, it enhances robustness under attacks (Madry et al., 2018; Zhang et al., 2019; Dong et al., 2024d;b). In this paper, we extend adversarial robustness in the context of multi-modal zero-shot generalization.

**Multi-modal adversarial robustness.** The computational cost impedes scaling to large VLMs such as CLIP (Radford et al., 2021). To address this limitation, adversarial fine-tuning (Mao et al., 2023), often leveraging Parameter-Efficient Fine-tuning (PEFT) (Jia et al., 2022; Zhou et al., 2022; Ni et al., 2024; Zhu et al., 2025), has attracted increasing attention. Mao et al. (2023) introduced adversarial fine-tuning through text-guided contrastive learning, aligning image-text embeddings for adversarial robustness. To mitigate the potential over-fitting to fine-tuning datasets, Wang et al. (2024) designed a prediction-level regularization guided by natural CLIP, while Schlarmann et al. (2024) proposed an unsupervised adversarial framework. Despite their efficacy, existing approaches adopt a point-wise align-

ment that integrates a single adversarial counterpart per clean sample, overlooking the broader spectrum of plausible adversaries in the vicinity of the decision boundary, thus compromising zero-shot robustness against unforeseen adversaries. In contrast, we "incorporate" entire "adversarial simplices" into the robustification process.

## 5. Experiments

Below, we provide our experimental configurations and present our comparisons between our *AdvSimplex* and other adversarial fine-tuning models across 15 datasets.

**Datasets.** We adopt the setup from Mao et al. (2023); Wang et al. (2024); Schlarmann et al. (2024), where CLIP is adversarially fine-tuned on the ImageNet training set (Deng et al., 2009). Then we assess the zero-shot results of the fine-tuned CLIP on the ImageNet val set and 14 novel datasets. We also investigate our method in medical image analysis and vision-text understanding. See settings in Appendix B.1.

**Implementation details.** Unless specified otherwise, we use CLIP (Radford et al., 2021) based on ViT-Base/32 (Dosovitskiy et al., 2021), as per studies (Mao et al., 2023; Wang et al., 2024; Schlarmann et al., 2024). For adversary generation during fine-tuning, we employ PGD (Madry et al., 2018) with $m = 10$ iterations under the $\ell_\infty$-norm threat model, the perturbation radius $\epsilon = 2/255$ and the step size $\alpha = 1/255$. The weighting factor is set to $\lambda = 0.6$. During evaluations, we assess the natural and robust performance under three strong white-box adversarial attacks: PGD (Madry et al., 2018) with 20 iterations, CW (Carlini

*Table 4.* Comparison of our method with other approaches, reporting average accuracy (%) across **diverse CLIP architectures**.

| Architecture | Method | Clean | PGD | CW | AA |
|---|---|---|---|---|---|
| ViT-B | TeCoA | 48.83 | 27.33 | 26.80 | 25.75 |
| | PMG-FT | 49.71 | 28.44 | 27.63 | 26.98 |
| | FARE | 56.68 | 30.94 | 30.26 | 29.30 |
| | *AdvSimplex* | **60.23** | **35.68** | **34.93** | **34.06** |
| ViT-L | TeCoA | 70.95 | 54.19 | 50.84 | 47.96 |
| | PMG-FT | 68.67 | 55.75 | 52.29 | 49.38 |
| | FARE | 71.30 | 57.12 | 54.16 | 50.54 |
| | *AdvSimplex* | **73.39** | **59.52** | **57.64** | **52.80** |
| ResNet-50 | TeCoA | 43.25 | 23.20 | 22.53 | 21.74 |
| | PMG-FT | 45.11 | 24.68 | 23.72 | 23.05 |
| | FARE | 45.43 | 24.27 | 23.19 | 22.24 |
| | *AdvSimplex* | **48.85** | **26.23** | **25.38** | **24.70** |

*Table 5.* Comparison of our method with other adv. fine-tuning approaches. Average accuracy (%) under **diverse $\epsilon$** are reported.

| Perturbation Radius | Method | PGD | CW | AA |
|---|---|---|---|---|
| $\epsilon_I = 1/255$ | TeCoA | 38.10 | 36.62 | 35.83 |
| | PMG-FT | 38.92 | 37.45 | 36.98 |
| | FARE | 45.40 | 44.81 | 43.27 |
| | *AdvSimplex* | **48.12** | **46.94** | **45.61** |
| $\epsilon_I = 2/255$ | TeCoA | 27.33 | 26.80 | 25.75 |
| | PMG-FT | 28.44 | 27.63 | 26.98 |
| | FARE | 30.94 | 30.26 | 29.30 |
| | *AdvSimplex* | **35.68** | **34.93** | **34.06** |
| $\epsilon_I = 3/255$ | TeCoA | 17.90 | 17.51 | 17.08 |
| | PMG-FT | 19.14 | 18.69 | 18.22 |
| | FARE | 19.31 | 18.84 | 18.47 |
| | *AdvSimplex* | **25.68** | **24.80** | **24.34** |
| $\epsilon_I = 4/255$ | TeCoA | 11.23 | 10.70 | 10.35 |
| | PMG-FT | 12.05 | 11.56 | 11.17 |
| | FARE | 12.42 | 11.93 | 11.52 |
| | *AdvSimplex* | **18.63** | **18.10** | **17.58** |

& Wagner, 2017), and Auto-Attack (AA) (Croce & Hein, 2020). All evaluations use adaptive attack schemes for fair comparison. See implementation details in Appendix B.2.

## 5.1. Main Results

**Evaluations across 15 datasets.** We compare our *AdvSimplex* with TeCoA (Mao et al., 2023), PMG-FT (Wang et al., 2024), and FARE (Schlarmann et al., 2024) in Tables 2 & 3, where we also provide zero-shot inference on additional 14 datasets, reporting natural performance and robustness against 20-step PGD attacks. *AdvSimplex* consistently outperforms other models in clean accuracy, with an average improvement of 3.5%, thus approaching standard CLIP (Table 2). While the standard CLIP has nearly zero adv. robustness (Table 3), our *AdvSimplex* enjoys an average improvement of 4.7% across all datasets in comparison to FARE.

**Adversarial fine-tuning of diverse architectures.** Table 4 reports zero-shot results on various clip architectures for clean samples and adversarial counterparts across three attack types ($\epsilon = 2/255$): PGD (Madry et al., 2018) (20 steps), CW (Carlini & Wagner, 2017), and Auto-Attack (Croce & Hein, 2020). Our *AdvSimplex* outperforms other adversarial fine-tuning models under all architectures.

*Table 6.* Comparison of our method with other approaches, reporting average accuracy (%) on **text-level and bi-level adversaries**.

| Method | Text-Level Attacks | | Bi-Level Attacks | |
|---|---|---|---|---|
| | BERT-Attack | GBDA | Co-Attack | SGA |
| TeCoA | 37.14 | 35.30 | 26.73 | 25.94 |
| PMG-FT | 37.61 | 36.46 | 28.11 | 27.85 |
| FARE | 35.45 | 34.97 | 25.38 | 25.06 |
| *AdvSimplex* | **40.21** | **39.88** | **32.95** | **32.53** |

*Table 7.* Average accuracy (%) using adversaries of varying $\epsilon$ for both fine-tuning and evaluations (ViT-B) with **the VPT strategy**.

| Perturbation Radius | Method | Clean | PGD | CW | AA |
|---|---|---|---|---|---|
| $\epsilon_I = 1/255$ | TeCoA | 51.00 | 32.27 | 31.11 | 30.26 |
| | PMG-FT | 52.64 | 33.09 | 32.10 | 30.83 |
| | FARE | 52.75 | 32.69 | 31.58 | 30.64 |
| | *AdvSimplex* | **55.08** | **34.96** | **33.75** | **33.16** |
| $\epsilon_I = 2/255$ | TeCoA | 42.61 | 18.12 | 16.88 | 15.39 |
| | PMG-FT | 42.11 | 19.26 | 17.68 | 16.47 |
| | FARE | 42.81 | 18.98 | 17.46 | 16.35 |
| | *AdvSimplex* | **44.70** | **22.57** | **21.13** | **20.28** |
| $\epsilon_I = 3/255$ | TeCoA | 33.86 | 12.32 | 10.78 | 8.89 |
| | PMG-FT | 32.52 | 12.87 | 11.36 | 9.38 |
| | FARE | 33.70 | 12.47 | 10.92 | 9.04 |
| | *AdvSimplex* | **35.11** | **15.38** | **14.42** | **11.96** |
| $\epsilon_I = 4/255$ | TeCoA | 26.78 | 11.04 | 9.87 | 7.19 |
| | PMG-FT | 23.57 | 11.73 | 10.01 | 7.26 |
| | FARE | 26.17 | 11.49 | 10.32 | 7.53 |
| | *AdvSimplex* | **29.23** | **14.25** | **12.63** | **10.22** |

**Adversarial fine-tuning w.r.t. diverse perturbation radii.** Table 5 shows the average clean and robust accuracy across 15 datasets w.r.t. various perturbation radii. Our *AdvSimplex* consistently surpasses other adversarial fine-tuning methods when facing stronger adversaries in zero-shot settings.

**Robustness on text-level and bi-level attacks.** In addition to image-level attacks, we assess text-level and bi-level attacks. Table 6 reports robust accuracy under these settings. Text-level attacks are evaluated using BERT-Attack (Li et al., 2020) and Gradient-Based Distributional Attack (GBDA) (Guo et al., 2021), while bi-level adversaries are tested using Collaborative Multi-modal Adversarial Attack (Co-Attack) (Zhang et al., 2022a) and Set-level Guidance Attack (SGA) (Lu et al., 2023). Our *AdvSimplex* outperforms existing adversarial fine-tuning methods on all attack types.

**Efficient fine-tuning with VPT.** Fine-tuning the full parameter space is computationally expensive for VLMs. Thus, we investigate adversarial fine-tuning with Visual Prompt Tuning (VPT) (Jia et al., 2022), a parameter-efficient strategy with learnable parameters in the token embedding layer. Table 7 shows the zero-shot performance of *AdvSimplex* under various adversarial configurations, comparing it with other techniques that also use VPT. Our *AdvSimplex*, even when using VPT for efficiency, outperforms previous works.

## 5.2. Extensions to Other Architectures and Tasks

**BLIP: Vision-text Understanding.** Beyond standard CLIP (Radford et al., 2021), we further explore zero-shot robust-

*Table 8.* BLIP Extension for Vision-Text Understanding. Evaluations on clean and PGD-20 adversarial samples. TR and IR represent the recall@1 for text and image retrieval, respectively. CIDEr measures the similarity of a generated sentence against a set of ground truth sentences for image captioning evaluations.

| Method | Image-Text Retrieval | | | | Image Captioning | |
|---|---|---|---|---|---|---|
| | Clean TR | Robust TR | Clean IR | Robust IR | Clean CIDEr | Robust CIDEr |
| TeCoA | 87.5 | 54.4 | 77.0 | 47.5 | 96.9 | 57.8 |
| PMG-FT | 87.8 | 55.6 | 77.9 | 48.2 | 97.5 | 58.2 |
| FARE | 88.2 | 55.9 | 78.4 | 49.0 | 98.1 | 58.7 |
| *AdvSimplex* | **91.7** | **58.7** | **80.9** | **51.6** | **99.5** | **63.4** |

*Table 9.* Medical CLIP Extension for Diagnosis. AUC score evaluations on both clean and PGD-20 adversarial samples.

| Method | ChestXray14 | | CheXpert | | PadChest | |
|---|---|---|---|---|---|---|
| | Clean | PGD | Clean | PGD | Clean | PGD |
| TeCoA | 0.674 | 0.526 | 0.857 | 0.685 | 0.602 | 0.483 |
| PMG-FT | 0.692 | 0.538 | 0.850 | 0.688 | 0.619 | 0.495 |
| FARE | 0.687 | 0.533 | 0.845 | 0.679 | 0.615 | 0.490 |
| *AdvSimplex* | **0.742** | **0.579** | **0.880** | **0.735** | **0.632** | **0.563** |

ness under alternative VLM architectures and downstream tasks using BLIP (Li et al., 2022), which combines multiple vision-language understanding tasks. We consider two cross-modal tasks: (i) image-text retrieval using Flickr30k (Plummer et al., 2015), and (ii) image captioning using Nocaps (Agrawal et al., 2019). For adversarial fine-tuning, we adversarially optimize the Image-Text Contrastive (ITC) learning, Image-Text Matching (ITM), and Language Modeling (LM) modules (Li et al., 2021), instead of performing alignment as in Eq. (3). Table 8 shows zero-shot results on clean samples and PGD-based adversarial attacks (20 steps) with $\epsilon = 1/255$, where *AdvSimplex* enjoys great adaptability.

**Medical CLIP: Medical Diagnosis.** Below, we investigate *AdvSimplex* in the medical imaging domain, where adversarial threats may affect computer-aided diagnostics (Zhao et al., 2023b). We employ a radiology-oriented CLIP variant under the CheXzero paradigm (Tiu et al., 2022) with a ViT-B backbone, again applying adversarial fine-tuning. We measure zero-shot performance on three multi-label radiology benchmarks: ChestX-ray14 (Wang et al., 2017), CheXpert (Irvin et al., 2019), and PadChest (Bustos et al., 2020). We report the Area Under the Curve (AUC) on clean and adversarial samples, where adversaries are generated via 20-step PGD under $\epsilon = 1/255$. Table 9 shows that our method consistently obtains higher AUC scores than prior approaches in clean and adversarial samples. Our method enjoys superior robustness on PadChest, which includes 192 disease categories and numerous uncommon pathologies.

## 5.3. Further Analyses

Below, we analyze the effectiveness and generalizability of our *AdvSimplex* across diverse settings.

*Table 10.* Average clean and robust accuracy (%) of our *AdvSimplex* method *vs. AdvTetrahedron* across 15 datasets.

| Configuration | Clean | PGD | AA |
|---|---|---|---|
| *AdvSimplex* | 60.23 | 35.68 | 34.06 |
| *AdvTetrahedron* | **60.80** | **36.49** | **35.17** |

*Table 11.* Average Performance (%) of our *AdvSimplex* method using different configurations of the derived upper bound $\Omega$.

| Configuration | Clean | PGD | AA |
|---|---|---|---|
| w/o cross-product term ($\bar{\bar{\Omega}}$) | 59.97 | 34.92 | 33.19 |
| w/ cross-product term ($\bar{\Omega}$) | **60.23** | **35.68** | **34.06** |

*Table 12.* Ablation study of three components in our method for average clean and robust accuracy (%) on 15 datasets.

| | $\bar{\bar{\Omega}}$ | Re-weighting (Eq. (16)) | Clean | PGD | CW | AA |
|---|---|---|---|---|---|---|
| 1 | | | 55.79 | 30.24 | 29.50 | 28.86 |
| 2 | ✓ | | 59.45 | 34.38 | 33.64 | 32.96 |
| 3 | | ✓ | 56.92 | 33.17 | 32.49 | 31.72 |
| 4 | ✓ | ✓ | **60.23** | **35.68** | **34.93** | **34.06** |

**Higher-order simplices.** According to Theorem 3.1 for simplices with $Q > 3$ vertices, *i.e.*, a tetrahedron ($Q = 4$ vertices), Table 10 shows gains for $(\mathbf{x}, \mathbf{x}+\boldsymbol{\delta}_{x,i}, \mathbf{x}+\boldsymbol{\delta}_{x,i+1}, \mathbf{x}+\boldsymbol{\delta}_{x,i+2})$ for consecutive $i = 1, \ldots, m - 2$ simplices from gradient ascent steps over *AdvSimplex* ($Q = 3$ vertices).

**Cross-product term in the upper bound $\bar{\Omega}(\mathbf{x})$.** Recall that we have a cross-product term $\gamma$ (last term in approx. $\Omega(\mathbf{x})$ in Eq. (6)) for a better estimation of the upper bound of the prediction gap. Table 11 analyzes its impact on adversarial fine-tuning. A more precise estimation with the cross-product term improves the zero-shot adversarial robustness. The cross-term product is derived using Kronecker operations, resulting in third-order statistics. Such statistics also have a closed-form solution, and the dimensionality of tensors is tensor-sketched to keep calculations fast (Weinberger et al., 2009). See Appendix C.3 for derivations.

**Impact of each module.** Below we ablate two key components of *AdvSimplex*: (i) $\bar{\bar{\Omega}}$, and (ii) weighting in Eq. (16). Table 12 shows that our baseline (first row) follows the surrogate optimization of robust risk (*i.e.*, TRADES (Zhang et al., 2019)) by extending the point-wise clean-adversarial prediction alignment with intermediate adversaries. Despite its simplification, our baseline approach already achieves competitive performance compared to prior adversarial fine-tuning methods. Incorporating "adversarial simplices" yields further gains in both clean and robust accuracy. Our adaptive re-weighting emphasizes that not every simplex is equally adversarial.

**Accuracy-robustness trade-off.** Striking a balance between natural performance and adversarial robustness is known from uni-modal adversarial learning (Zhang et al., 2019; Dong et al., 2023a). Below, we explore it in the multi-

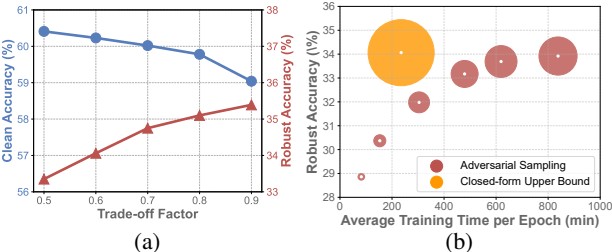

**(a)**      **(b)**

*Figure 4.* Evaluations of the average performance (%) across 15 datasets. (a) Trade-off between zero-shot clean and robust accuracy by tuning $\lambda$. (b) Comparison of our closed-form upper bound with adversarial sampling at varying sample sizes (the circle radius indicates the number of samples; for "closed-form" radius is $\infty$).

*Table 13.* Average Performance (%) of our proposed method with diverse re-weighting strategies for adversarial triangular regions.

| Re-Weighting Strategy | Clean | PGD | AA |
|---|---|---|---|
| Uniform Weighting (No Weights) | 59.45 | 34.38 | 32.96 |
| Linear Weighting ($i/m$) | 58.74 | 35.12 | 33.48 |
| Adaptive Weighting (**Ours**) | **60.23** | **35.68** | **34.06** |

modal CLIP and zero-shot scenario. We analyze the effect of hyper-parameter $\lambda$, which controls the relative weighting of clean sample classification *vs.* adversarial-clean prediction alignment. Figure 4a shows that increasing $\lambda$ enhances adversarial robustness yet reduces clean accuracy. Conversely, lowering $\lambda$ improves zero-shot performance on benign inputs at the cost of reduced robustness. Such a trade-off stems from the optimization of natural and boundary risks.

**Performance of sampling *vs*. closed-form solution.** Recall that we introduce an efficient upper bound derived with a closed-form solution replacing sampling "adversarial simplices". Below, we compare our "upper-bound closed-form model" against explicit sampling of "adversarial simplices" for the standard alignment loss. Figure 4b shows that although increasing the sampling amount leads to a gradual improvement in adversarial robustness, it stabilizes around 70 samples and saturates around 100 samples, posing substantial computational training time of 13.6 hours per epoch. In contrast, our *AdvSimplex* requires merely 4.1 hours per epoch while attaining comparable robustness.

**Re-weighting mechanisms for "adv. simplices".** Below we compare the use of weights from Eq. (16) with (i) uniform weighting *vs*. (2) linear weighting ($i/m$) that place greater emphasis on adversaries from later iteration steps. Table 13 shows that our Eq. (16) outperforms other variants.

**Closed-form *vs*. sampled "adversarial simplex".** Below, we analyze the average robust accuracy of the closed-form "adversarial simplices" *vs*. sampled "adversarial simplices" (70 samples per simplex). We attack both methods with an index $i$ adversary from PGD-20. Table 14 shows that the closed-form solution enjoys greater zero-shot robustness

*Table 14.* Average robust accuracy (%) for closed-form "adversarial simplices" *vs.* sampling from "adversarial simplices" (70 samples per simplex) evaluated on attack samples with index number $i = 6, 8, 10, 12, 14$ generated via PGD-20.

| Optimization Strategy | Intermediate Adv. Sample Index (Attack) | | | | |
|---|---|---|---|---|---|
| | 6-th | 8-th | 10-th | 12-th | 14-th |
| Sampled simplex | 35.66 | 35.18 | 34.84 | 34.47 | 34.19 |
| **Closed-form simplex** | **38.61** | **37.89** | **37.22** | **36.50** | **36.13** |

*Table 15.* Average robust accuracy (%) on worst-case and most transferable adversaries sampled from "adversarial simplices".

| Optimization Strategy | $\epsilon = 2/255$ | | | $\epsilon = 4/255$ | | |
|---|---|---|---|---|---|---|
| | PGD | Worst-Case | Transfer | PGD | Worst-Case | Transfer |
| Sampled simplex | 35.19 | 36.86 | 40.90 | 16.92 | 21.15 | 33.28 |
| **Closed-form simplex** | **35.68** | **39.51** | **43.38** | **18.63** | **23.72** | **35.92** |

against adversarial attacks of various step numbers.

**Performance w.r.t. sampled "adversarial simplices".** In addition to attacks along the generation path, we also evaluate the adversarial robustness against both the worst-case and the most transferable adversaries sampled from "adversarial simplices". Following the setup from Figure 3, worst-case adversaries are obtained from the target CLIP model, and the most transferable adversaries from three other CLIP models. Table 15 shows that our derived closed-form upper-bound model enjoys greater robustness against adversaries from "adversarial simplices".

## 6. Conclusion

Motivated by our analysis of the robustness degradation against underlying adversaries from "adversarial simplices", we have uncovered that the point-wise prediction alignment in robust VLMs leads to weak robustness generalization. Thus, we have explored recent attack strategies to formulate simplices between clean vertex $\mathbf{x}$ and consecutive adversarial samples on the gradient ascent path. While sampling such simplices is prohibitive, and aligning such adversarial candidate points is also prohibitive, one may reformulate the problem by minimizing an upper bound of the alignment loss. Our upper bound employs closed-form statistics obtained from the vertices of simplices, the Jacobian and Hessian matrices. We only pass clean samples via the encoder, reducing time complexity, and we achieve "infinite sampling" effect with our formulation during fine-tuning.

## Acknowledgements

This research is supported by National Research Foundation, Singapore and Infocomm Media Development Authority under its Trust Tech Funding Initiative, the Centre for Frontier Artificial Intelligence Research, Institute of High Performance Computing, A*Star, and the College of Computing and Data Science at Nanyang Technological University.

Any opinions, findings, conclusions, or recommendations expressed in this material are those of the author(s) and do not reflect the views of National Research Foundation, Singapore, and Infocomm Media Development Authority. Piotr Koniusz and Hao Zhu are supported by CSIRO's Science Digital.

## Impact Statement

The advancement of Vision-Language Models (VLMs) has revolutionized zero-shot classification and a series of vision-language understanding tasks. However, the deployment of these models in real-world applications necessitates robust defenses against (unforeseen) adversarial attacks, which can significantly undermine their performance and trustworthiness. Our proposed adversarial fine-tuning method addresses this critical security vulnerability by enhancing the robustness of VLMs against adversarial samples, ensuring more reliable performance in diverse scenarios. The improvement in zero-shot adversarial robustness can further lead to several positive outcomes as follows:

1. **Research Advancement**: Our findings on the correlation between adversaries on simplex and adversarial robustness provide insights for the broader research community related to robust cross-modal learning.

2. **Enhanced AI Safety**: By improving the robustness of foundational VLMs against unforeseen adversarial attacks, our method contributes to the development of more secure AI systems that can be safely deployed in sensitive applications, such as autonomous driving, healthcare diagnostics, and security surveillance.

3. **Social Trust**: As AI systems become increasingly integrated into daily life, ensuring their reliability and robustness is important. Our improvement of adversarial robustness on foundational models further helps build societal trust in AI technologies.

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

# Appendices.

## A. Limitations.

Although our method yields substantial gains in zero-shot adversarial robustness, supported by both empirical results and theoretical analysis, we acknowledge two primary limitations that suggest avenues for future improvements:

1. **Computational Overhead**: While our adversary-generation strategy and subsequent optimization via the closed-form upper save time during fine-tuning over naive sampling of simplex, this cost would still benefit from further reduction. It is worth noting that the inference stage does not incur extra overhead relative to previous methods.

2. **Implicit Trade-off Using PEFT**: In this work, we primarily concentrate on adversarial full fine-tuning of vision-language models. According to Tables 5 and 7, it is evident that although Parameter-Efficient Fine-Tuning (PEFT) provides improved training efficiency, it can also degrade zero-shot performance on both clean and adversarial data compared to full fine-tuning. This performance-efficiency trade-off affects all adversarial fine-tuning methods, and we anticipate that future advancements in PEFT will help minimize the robustness gap compared to full fine-tuning.

3. **Positive Applications of Adversarial Examples**: While our work primarily focuses on defending against adversarial examples and mitigating their associated threats, adversarial examples can also serve constructive purposes. For instance, they are widely used as robustness benchmarks to evaluate model reliability under distribution shifts (Carlini et al., 2019; Croce et al., 2021), and as tools for privacy protection (Dong & Xie, 2021; Dong et al., 2023b; Chen et al., 2025). A more comprehensive treatment of these positive applications is beyond the scope of this work, but we consider it an important direction for future research.

## B. Experimental Configurations.

In this section, we provide a comprehensive overview of the experimental configuration used throughout our work, including both dataset specifications for adversarial fine-tuning/evaluations and the implementation details of our proposed method.

### B.1. Dataset descriptions

We follow prior works (Mao et al., 2023; Wang et al., 2024) to adversarially fine-tune the CLIP model on the ImageNet training split (Deng et al., 2009) and evaluate its performance on the ImageNet validation set, as the ground-truth labels for the test set are not publicly available. Additionally, we assess the CLIP model on 14 diverse zero-shot datasets, covering a broad spectrum of image recognition tasks. Collectively, these 15 datasets encompass:

- **General Image Classification:** ImageNet (Deng et al., 2009), STL-10 (Coates et al., 2011), CIFAR-10/100 (Krizhevsky et al., 2009), Caltech-101 (Fei-Fei et al., 2004), and Caltech-256 (Griffin et al., 2007).

- **Fine-Grained Classification:** FGVC Aircraft (Maji et al., 2013), Flower102 (Nilsback & Zisserman, 2008), Food101 (Bossard et al., 2014), Oxford-IIIT Pets (Parkhi et al., 2012), and Stanford Cars (Krause et al., 2013).

- **Domain-Specific Classification:** Describable Textures Dataset (DTD) (Cimpoi et al., 2014), EuroSAT (Helber et al., 2019), and PatchCamelyon (PCAM) (Veeling et al., 2018).

- **Scene Recognition:** SUN397 (Xiao et al., 2010).

We use standard data pre-processing (image resizing to the resolution of $224 \times 224$ and center-cropping) during adversarial fine-tuning to ensure the consistency with prior works (Mao et al., 2023; Wang et al., 2024; Schlarmann et al., 2024). In addition to the datasets used for zero-shot classification, we further incorporate vision-language understanding and medical imaging datasets, with additional details provided in the following section.

### B.2. Implementation details

**Standard Setup.** Following established adversarial fine-tuning works (Mao et al., 2023; Wang et al., 2024), we adopt the CLIP model (Radford et al., 2021) with the ViT-Base/32 backbone (Dosovitskiy et al., 2021). We use the SGD optimizer

with a momentum factor of $0.9$, a batch size of $512$, and an initial learning rate of $1 \times 10^{-5}$ (scheduled via cosine decay) to optimize the image encoder of the CLIP model. For Parameter-Efficient Fine-Tuning (PEFT) with Visual Prompt Tuning (VPT) (Jia et al., 2022), we insert learnable tokens of size $100$ into the vision branch of CLIP and use a learning rate of $40$. For adversary generation during the fine-tuning stage, we employ PGD (Madry et al., 2018) with $m = 10$ iterations under the $\ell_\infty$-norm threat model, setting the perturbation radius $\epsilon = 2/255$ and the step size $\alpha = 1/255$, unless stated otherwise. The weighting factor is set to $\lambda = 3.0$. All robustness evaluations are performed using adaptive attack schemes to ensure a fair comparison. All the experiments were run on eight NVIDIA H100 GPUs.

**Evaluation Protocol.** In line with prior studies on adversarially robust CLIP (Mao et al., 2023; Wang et al., 2024; Schlarmann et al., 2024), we evaluate our method with both clean samples and three strong white-box adversaries: 20-step PGD (Madry et al., 2018), the CW attack (Carlini & Wagner, 2017), and Auto-Attack (AA) (Croce & Hein, 2020), focusing mainly on adversarial images to align with real-world defense applications. We additionally assess robustness against text-level adversaries (BERT-Attack (Li et al., 2020) and GBDA (Guo et al., 2021)) and bi-level adversaries from both image and text branches (Co-Attack (Zhang et al., 2022a) and SGA (Lu et al., 2023)), as described in the main text.

**BLIP Extensions.** To examine zero-shot robustness on downstream vision-language understanding tasks, we incorporate the BLIP framework (Li et al., 2022), which unifies vision-language tasks via bootstrapped pre-training. Specifically, we consider two cross-modal tasks: (i) image-text retrieval using the Flickr30k dataset (Plummer et al., 2015), and (ii) image captioning using the Nocaps dataset (Agrawal et al., 2019). We conduct adversarial min-max optimization of the Image-Text Contrastive (ITC), Image-Text Matching (ITM), and Language Modeling (LM) objectives following (Li et al., 2021) with our approach as an addition optimization term to obtain a robust version of BLIP. We then evaluate its robustness using iterative PGD for both image-text retrieval (maximizing the ITM loss) and image captioning (maximizing the LM loss). We generate and evaluate adversarial samples using the perturbation configuration of $\epsilon = 1/255$.

**Medical CLIP Extensions.** Beyond evaluations on natural images, we further investigate robustness in medical imaging using a specialized CLIP model (ViT-B/16) trained via CheXzero (Tiu et al., 2022) on chest radiographs and medical reports from the MIMIC database (Johnson et al., 2019). The text encoder of the CLIP model is based on BioBERT (Lee et al., 2020), tailored for biomedical language. We evaluate zero-shot AUC on ChestX-ray14 (Wang et al., 2017), CheXpert (Irvin et al., 2019), and PadChest (Bustos et al., 2020) under 20-step PGD attacks with the perturbation radius of $\epsilon = 1/255$.

## C. Further Analyses

### C.1. Derivation of bound in Eq. (7)

In Eq. (5) & (6), elements after the sum can be written as $(\sqrt{\alpha} + \frac{1}{2}\sqrt{\beta})^2 = \alpha + \frac{1}{4}\beta + 2 \cdot \frac{1}{2} \cdot \gamma$, where $\gamma = \sqrt{\alpha\beta}$ and $\alpha, \beta \geq 0$. Using the known inequality $(a + b)^p \leq 2^{p-1}(a^p + b^p)$ and setting $p = 2$, $a = \sqrt{\alpha}$, $b = \frac{1}{2}\sqrt{\beta}$, we obtain $(\sqrt{\alpha} + \frac{1}{2}\sqrt{\beta})^2 \leq 2\alpha + \frac{1}{2}\beta$. This upper bound eliminates the $\gamma$ term, which is why it does not appear in Eq. (7). This result corresponds to $\bar{\bar{\Omega}}$ in Eq. (11). For completeness, we evaluate the term with $\gamma$ in Appendix C.3, but it is computationally expensive due to operating in $\mathbb{R}^{(wh)^2}$ instead of $\mathbb{R}^{wh}$.

### C.2. Hyper-parameter analyses

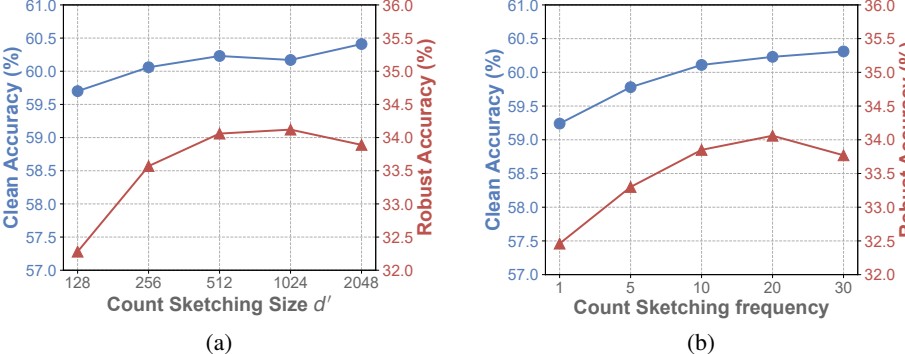

(a)                                                                  (b)

*Figure 5.* Hyper-parameter (Count sketching size $d'$ and refreshment frequency of count sketching matrices) sensitivity of our adversarial fine-tuning method on average clean and (Auto-Attack) robust accuracy (%) across 15 datasets in the zero-shot setting.

Beyond our analysis of the loss weighting factor $\lambda$ described in Figure 4a in the main text, we further analyze how other hyper-parameters influence our adversarial fine-tuning method, especially using diverse configurations of the count sketching for accelerating the inner product computation.

As shown in Figure 5, we present both clean accuracy and (Auto-Attack) robust accuracy under various hyper-parameter configurations. Note that the setting of all the hyper-parameters is obtained through the Hyperopt package (Bergstra et al., 2013) for a 25-iteration hyper-parameter search on a 1% subset of the ImageNet training set to ensure consistency and was subsequently applied across diverse adversarial fine-tuning scenarios. Notably, appropriate choices of the count sketching hyper-parameters enable a balanced trade-off between natural performance and adversarial robustness in the zero-shot setting.

### C.3. The cross-product term of the upper bound $\bar{\Omega}(\mathbf{x})$.

The aggregation over the cross-terms $\gamma(\mathbf{x}, \boldsymbol{\delta}_x)$ can be expanded as follows:

$$\frac{1}{\kappa} \sum_{\boldsymbol{\delta}_x \in \Delta_{\mathcal{X}}} \left\langle J_g(\mathbf{x}) \boldsymbol{\delta}_x, \left[\boldsymbol{\delta}_x^T (H_g(\mathbf{x}))_c \boldsymbol{\delta}_x\right]_{c=1}^C \right\rangle = \frac{1}{\kappa} \sum_{\boldsymbol{\delta}_x \in \Delta_{\mathcal{X}}} \sum_{c=1}^C \left\langle \boldsymbol{\delta}_x, J_g(\mathbf{x}))_{c,:}^\top \right\rangle \left\langle \boldsymbol{\delta}_x \boldsymbol{\delta}_x^\top, (H_g(\mathbf{x}))_c \right\rangle \tag{25}$$

$$= \left\langle \frac{1}{\kappa} \sum_{\boldsymbol{\delta}_x \in \Delta_{\mathcal{X}}} \boldsymbol{\delta}_x \otimes \mathrm{vec}(\boldsymbol{\delta}_x \boldsymbol{\delta}_x^\top), \sum_{c=1}^C J_g(\mathbf{x}))_{c,:}^\top \otimes \mathrm{vec}((H_g(\mathbf{x}))_c) \right\rangle \tag{26}$$

$$\approx \left\langle \frac{1}{\kappa} \sum_{\boldsymbol{\delta}_x \in \Delta_{\mathcal{X}}} \mathbf{P}\boldsymbol{\delta}_x \otimes \mathrm{vec}(\mathbf{P}\boldsymbol{\delta}_x \boldsymbol{\delta}_x^\top \mathbf{P}^\top), \sum_{c=1}^C \mathbf{P}J_g(\mathbf{x}))_{c,:}^\top \otimes \mathrm{vec}(\mathbf{P}(H_g(\mathbf{x}))_c \mathbf{P}^\top) \right\rangle, \tag{27}$$

where $\mathbf{P} \in \mathbb{R}^{w'h' \times wh}$ is the projection matrix used by the count sketching, and $w'h' \ll wh$.

### C.4. Set-level augmentation variant of *AdvSimplex*

Recall that SGA (Lu et al., 2023) introduces adversarial perturbations in every adversary generation step (gradient ascent step) around the intermediate adversary $\mathbf{v}'$, which continues the ascent step from their mean ($\boldsymbol{\mu} \neq \mathbf{v}'$). Similar strategies have also been demonstrated to be effective in enhancing adversarial robustness (Addepalli et al., 2022; Dong et al., 2023c; 2024e). Thus, for the same $\mathbf{x}$, it is possible to generate three different adversarial trajectories. Let us consider an SGA variant of our *AdvSimplex* by using the simplex formed from the final 3 adversaries of 3 adversarial paths per sample $\mathbf{x}$, dubbed as *AdvSimplex-SGA*. Furthermore, we consider using the final adversarial example generated via SGA and generating 3 adversarial paths as in *AdvSimplex-SGA* and use the final 3 adversaries directly for robustification. The average results across 15 datasets are shown in Table 16.

Table 16. Average Performance (%) of our *AdvSimplex* method with the set-level augmentation.

| Configuration | Clean | AA |
|---|---|---|
| AT-SGA (standard: one adversary per $\mathbf{x}$) | 56.84 | 30.08 |
| AT-SGA (3 adversaries per $\mathbf{x}$) | 57.31 | 30.93 |
| *AdvSimplex-SGA* (simplex on 3 Adversaries per $\mathbf{x}$) | 58.75 | 32.40 |
| ***AdvSimplex*** | **60.23** | **34.06** |

### C.5. KL-divergence alternative to the Euclidean distance-based alignment

Prediction alignment via the KL-divergence has been widely used in the context of adversarially robust learning in previous studies (Zhang et al., 2019; Dong et al., 2022). Thus, in this section, we also consider a KL-divergence alternative to our *AdvSimplex* which does not use the Euclidean distance. Specifically, we replace $\Omega(\mathbf{x})$ by using a different Taylor expansion $\log g(\mathbf{x} + \boldsymbol{\delta}_\mathbf{x}) - \log g(\mathbf{x}) \approx J_{\log(g+\rho)}(\mathbf{x})\boldsymbol{\delta}_\mathbf{x} + \frac{1}{2}[\boldsymbol{\delta}_\mathbf{x}^\top (H_{\log(g+\rho)}(\mathbf{x}))_c \boldsymbol{\delta}_\mathbf{x}]_{c=1}^C$, where $\rho = 1 \times 10^{-5}$ is added for the numerical stability of log. Hence, we can obtain $\Omega_{\mathrm{KL}}(\mathbf{x}) = \frac{1}{\kappa} \sum_{\boldsymbol{\delta}_\mathbf{x} \in \Delta_{\mathcal{X}}} KL(g(\mathbf{x}) \| g(\mathbf{x} + \boldsymbol{\delta}_\mathbf{x})) \approx -\sum_{c=1}^C (g(\mathbf{x}))_c \cdot \left[\langle \boldsymbol{\mu}_\mathbf{x}, J_{\log(g+\rho)}(\mathbf{x}, c)\rangle + \frac{1}{2}\langle \boldsymbol{\Sigma}_\mathbf{x}, (H_{\log(g+\rho)}(\mathbf{x}))_c\rangle\right]$, where $\boldsymbol{\mu}_\mathbf{x}$ is the analytical mean of simplex, and $\boldsymbol{\Sigma}_\mathbf{x}$ is from Theorem

3.1. The average results across 15 datasets are shown in Table 17.

Table 17. Average Performance (%) of our *AdvSimplex* built with the KL-divergence.

| Configuration | Clean | AA |
|---|---|---|
| KL-divergence in Eq. (1) | 59.28 | 32.96 |
| *AdvSimplex-KL* (using $\Omega_{KL}(\mathbf{x})$ instead of $\Omega(\mathbf{x})$) | 59.13 | 33.20 |
| ***AdvSimplex*** | **60.23** | **34.06** |

## C.6. Comparison with TGA-ZSR

TGA-ZSR (Yu et al., 2024) designs text-guided attention to enhance zero-shot robustness of VLMs. Below we evaluate and compare our *AdvSimplex* with TGA-ZSR on image-level, text-level, and bi-level adversarial attacks. The average results across 15 datasets are shown in Table 18.

Table 18. Average Performance (%) of our *AdvSimplex* method *vs.* TGA-ZSR.

| Method | Image-level Attacks | | | Text-level Attacks | | Bi-level Attacks | |
|---|---|---|---|---|---|---|---|
| | Clean | PGD | AA | BERT-Attack | GBDA | Co-Attack | SGA |
| TGA-ZSR | 57.54 | 31.15 | 30.41 | 38.07 | 37.32 | 29.30 | 28.58 |
| *AdvSimplex* | **60.23** | **35.68** | **34.06** | **40.21** | **39.88** | **32.95** | **32.53** |

## C.7. Analysis of intermediate adversaries along the adversary generation trajectory

As the gradient ascent optimization during adversary generation uses a fixed gradient step size and the decision boundary of VLMs is non-linear, the adversarial path is thus not a perfect ascent. Specifically, intermediate adversarial samples along the trajectory may sometimes be stronger than final adversarial samples. In addition, these intermediate adversaries enjoy diversity in adversarial directions. To validate the dynamic weighting behavior described in Eq. (16), we empirically investigate whether the final-step adversary at iteration $m$ consistently receives a higher weight ($\omega_m$) than its intermediate counterparts. We note that the inequalities $\omega_m \geq \omega_{m-1}$, $\omega_{m-1} \geq \omega_{m-2}$, and $\omega_{m-2} \geq \omega_{m-3}$ hold only 83%, 79%, and 73% times on average, respectively. This observation supports our claim that the final-step adversary is not necessarily more adversarial than adversaries from earlier iterations, justifying the need for adaptive weighting across intermediate steps.

Inspired by the insights of Friendly Adversarial Training (Zhang et al., 2020), which emphasizes the role of weak adversaries in shaping robust decision boundaries, we analyze the impact of using simplices constructed from adversarial examples at different steps.

Table 19. Average Performance (%) of our *AdvSimplex* using simplices formed by subsets of intermediate adversary indices.

| Indices of Used Adversarial Simplices | Clean | AA |
|---|---|---|
| 1-3 (early steps) | 59.79 | 32.00 |
| 1-5 (early steps) | 60.37 | 32.49 |
| 6-10 (late steps) | 59.14 | 33.38 |
| 8-10 (late steps) | 58.65 | 33.06 |
| **1-10 (all)** | **60.23** | **34.06** |

Table 19 reveals that early adversarial steps (*e.g.*, steps 1–3) improve clean accuracy, likely due to their strong correlation with the original input and their augmentation effect. In contrast, late steps (*e.g.*, steps 8–10) contribute more to adversarial robustness. Notably, combining intermediate steps across the entire adversarial trajectory (1–10) achieves the best trade-off, yielding the highest clean and robust accuracy, suggesting that mid-level adversaries offer complementary benefits in balancing robustness and generalization.

