# OpenReview forum: "Improving Zero-Shot Adversarial Robustness in Vision-Language Models by Closed-form Alignment of Adversarial Path Simplices"
_ICML.cc/2025/Conference — ICML 2025 spotlightposter_

### Official Review · Reviewer_srgi · 2025-03-13

**Overall Recommendation:** 3

**Summary:**

In this work, the authors focus on a topic, robustness of VLM (CLIP). The authors propose to solve a challenge in adversarial fine-tuning, and propose to use Taylor expansion to enlarge the dicision space, which can make the model more robust.

**Claims And Evidence:**

The claims are supported by some math derivation.

**Essential References Not Discussed:**

The discussion of related works are sufficient.

**Experimental Designs Or Analyses:**

The work is all SOTA across all the tables of quantitative result, which is a large improvement.

**Methods And Evaluation Criteria:**

The paper uses a lot of quantitative results to demonstrate the effectiveness of this work.

**Other Comments Or Suggestions:**

I am willing to raise the score after effective rebuttal.

**Other Strengths And Weaknesses:**

Strength:

1. The quantitative results are a large improvement against baselines.

2. The figures are of high quality.

3. The paper provide theoretical analysis of the proposed theory.

Weakness:

Although the idea of the paper may be good, the writing of this work is not good, which make this work not very suitable for publication.

1. I think the writing of abstract needs to be improved. It should convey the key idea of the paper, instead of listing complex math notation. I suggest moving the math notation into the method section and just use natural language to describe the key idea of this work.

2. The related work should be in a seperate section instead of put in the introduction section. Also, the introduction is too long and too redundant.

3. The section 3 is too intensive with math. I think there should be some explanation of notations and some text for why you are writing this line of equation.

4. Small errors: Equation (10) is out of the space. Table 6 is overlapped with other tables.

**Questions For Authors:**

Will you open-source the code?

**Relation To Broader Scientific Literature:**

There is no need to discuss broader scientific literature.

**Theoretical Claims:**

This work includes one theorem (3.1), which is easy to understand.

---

> ### Author Rebuttal · Authors · 2025-03-31
>
> # Response to Rev. srgi
>
> We thank Rev. for the constructive feedback. **Kindly also note additional Theory/Results in Resp. to Rev. EyNB**.
>
> ### **1. Abstract.**
>
> Thank you. We have removed now math notations:
>
> ```
> Vision-Language Models (VLMs), e.g., CLIP, excel at zero-shot classification due to large-scale pre-training yet are vulnerable to adversarial examples. Adversarial fine-tuning robustifies zero-shot models by aligning prediction scores of individual adversaries with their clean counterparts, which typically overlooks intermediate adversarial samples along the adversarial trajectory crossing the decision boundary. Such intermediate adversaries and their vicinity offer informative representations of the decision surface, which can further be improved by sampling adversarial candidates from simplices formed by joining vertex with consecutive vertices on the adversarial trajectory. However, sampling simplices for adversaries is prohibitively costly. To train robust VLM, we overcome these limitations by Taylor expansion and formulating an upper-bound of alignment loss that depends on Jacobian/Hessian obtained at clean samples. As regions between clean and intermediate adversarial samples capture a larger decision landscape, we robustify VLM by plausible adversaries on simplices by our closed-form formulations equivalent to infinite uniform sampling of the simplex. We obtain state-of-the-art robustness across 15 datasets and diverse vision-language tasks.
> ```
>
> ###  **2. Separate Section for Related Works & Concise Intro.**
>
> **Absolutely. This is very easy to achieve.**
>
> We will provide a separate detailed related work section in the revised version, including:
> - alignment schemes (plus necessary equation).
> - single-modal adversarial attacks/robustness
> - multimodal adversarial attacks/robustness for a better understanding.
>
> ### **3. More Explanations of Section 3.**
>
> - We first discuss the upper bound of the prediction alignment between clean and adversarial examples derived by Taylor expansion in Section 3.1
> -  In Section 3.2, we propose a theorem to avoid empirical aggregating/sampling from adversarial simplices. In other words, we provide a closed-form solution for sampling from adversarial simplices.
> - The overall loss function is shown in Section 3.3.
> - We further demonstrate that our method also bounds the robust risk in Section 3.4.
>
> We will add the summary table of symbols:
>
> |Symbols|Explanations|
> |-|-|
> |$\mathbf{x}$|Clean example|
> |$\bf{\delta}_{\mathbf{x}}$|Adversarial perturbation|
> |$g_\theta(\cdot)$|Prediction of CLIP|
> |$J_{g}$|Jacobian matrix|
> |$H_{g}$|Hessian matrix|
> |$\Omega$|Vanilla upper bound of the Euclidean prediction distance|
> |$\bar{\Omega}$|Upper bound with the cross-product term|
> |$\bar{\bar{\Omega}}$|Upper bound without the cross-product term|
>
> ### **4. Formatting Issues.**
>
> We apologize. We will fix Eq. 10 and one table that jumped between pages breaking formatting.
>
> ### **5. Code Release.**
>
> **We stress that our code, model weights, and setup will be publicly available.**
>
> $\color{blue}\text{General Additional Results:}$
> ### **6. Additional results (ViT-B vs. ViT-L).**
>
> We extend our Table 3 (ImageNet), showing the average acc. across 15 datasets on ViT-B and ViT-L:
> |Architecture|Method|ImageNet Clean|ImageNet AA|Avg. Clean|Avg. AA|
> |-|-|-|-|-|-|
> |ViT-B|TeCoA|54.43|25.19|48.83|25.75|
> |ViT-B|PMG-FT|51.33|24.94|49.71|26.98|
> |ViT-B|FARE|50.94|23.78|56.68|29.30|
> |ViT-B|**AdvSimplex**|61.28|32.26|60.23|34.06|
> |ViT-L|TeCoA|73.61|61.14|70.95|47.96|
> |ViT-L|PMG-FT|73.92|60.63|68.67|49.38|
> |ViT-L|FARE|72.51|57.20|71.30|50.54|
> |ViT-L|**AdvSimplex**|76.87|64.09|73.39|52.80|
>
> ### **7. Additional results (Batch Size).**
>
> For completeness, we present results with two different batch sizes, **128 and 512** to show our approach performs best.  The average results across 15 datasets on ViT-B are below:
>
> |Batch Size|Method|Avg. Clean|Avg. AA|
> |-|-|-|-|
> |128|TeCoA|48.70|25.62|
> |128|PMG-FT|49.45|26.74|
> |128|FARE|56.22|28.95|
> |128|**AdvSimplex**|59.86|33.54|
> |512|TeCoA|48.83|25.75|
> |512|PMG-FT|49.71|26.98|
> |512|FARE|56.68|29.30|
> |512|**AdvSimplex**|60.23|34.06|
>
> Our method does not require a large batch size, but large batch size can enhance a bit all methods.
>
> ### **8. Additional results: Auto-Attack Results for Each Dataset.**
>
> The auto-attack results ($\epsilon$=2/255) of ViT-B corresponding to Table 2 is below:
>
> |Method|ImageNet|STL10|CIFAR-10|CIFAR-100|SUN397|Stanf.Cars|Food101|OxfordPet|Flower102|DTD|EuroSAT|FGVC|PCAM|Caltech101|Caltech256|Avearge|
> |-|-|-|-|-|-|-|-|-|-|-|-|-|-|-|-|-|
> |TeCoA|25.19|69.38|42.01|21.40|16.58|4.09|12.51|40.87|14.96|16.07|11.19|1.68|12.06|54.53|43.74|25.75|
> |PMG-FT|24.94|69.98|43.28|21.48|16.70|6.04|13.57|41.06|15.68|17.21|11.89|2.16|19.57|56.16|44.92|26.98|
> |FARE|23.78|75.40|49.76|28.35|16.29|8.32|17.04|44.61|17.03|18.39|8.58|2.25|21.41|60.14|48.16|29.30|
> |**AdvSimplex**|32.26|77.82|55.64|30.99|18.80|9.45|19.87|53.67|18.90|19.85|12.52|4.41|40.53|64.85|51.34|34.06|

---

> > ### Comment · Reviewer_srgi · 2025-04-05
> >
> > Thank you for your rebuttal! I hope the writing will be improved in the final version.

---

> > > ### Author Response · Authors · 2025-04-05
> > >
> > > Esteemed Reviewer,
> > > \
> > > \
> > > Thank you for your kind message, and valuable comments helping us improve and refine our manuscript. Meantime, if there is anything else we can answer or explain or discuss further, kindly do let us know.
> > > \
> > > \
> > > Rest assured, all requested improvements will be made in the final paper.
> > >
> > > Kind regards,
> > > \
> > > Authors

---

### Official Review · Reviewer_XcU1 · 2025-03-14

**Overall Recommendation:** 3

**Summary:**

This paper aims to enhance the robustness of CLIP for zero-shot image classification.
It emphasizes that existing defense methods often disregard intermediate adversarial samples along the trajectory, which are found to be beneficial in this study.
The proposed method, AdvSimplex, uses an efficent method to statistically align clean sample and adversarial samples from the trajectory, instead of sampling adversarial examples from the trajectory area and aligning them with a clean sample one by one.
Experiments on ImageNet demonstrate improvements in both accuracy and robustness.

**Claims And Evidence:**

- In the abstract, the authors claim, “We obtain state-of-the-art with 10× speed-up”.
    - However, the computational cost is never comparable to that of state-of-the-art methods.

- Adversarial training fundamentally involves solving a min-max optimization problem. It is unclear why using weaker adversarial examples from the trajectory would enhance robustness. The authors cite [Gao et al., 2024], whose work focuses on improving the transferability of adversarial examples, which involves sacrificing attack strength on the surrogate model. However, the relevance of transferable adversarial examples to adversarial training remains insufficiently validated, since using such weaker attack is suboptimal in min-max optimization.

[Gao et al. 2024] Gao, S., Jia, X., Ren, X., Tsang, I., and Guo, Q. Boosting transferability in vision-language attacks via diversifica- tion along the intersection region of adversarial trajectory. ECCV 2024

**Essential References Not Discussed:**

N/A

**Experimental Designs Or Analyses:**

Experimental designs are mostly aligned with existing work. However, there are some concerns, which are mentioned in "Methods And Evaluation Criteria."

**Methods And Evaluation Criteria:**

- The numbers are very different from those of existing papers. For example, in [Schlarmann et al. 2024], the clean zeros-shot accuracy on ImageNet is over 70% for all FARE, TeCoA, and original CLIP; however, Table 1 shows much lower numbers. Also, the robust accuracy is much lower. Why?
    - [Schlarmann et al. 2024] Schlarmann, Christian, et al. "Robust clip: Unsupervised adversarial fine-tuning of vision embeddings for robust large vision-language models." ICML 2024

- Table 2: The PGD-20 results as the main results are not convincing. Auto-Attack results are reported only for the average of datasets. This lacks information for readers.

**Other Comments Or Suggestions:**

- P.2, Line.90: “Lu et al. (2023) obtained intermediate adversarial samples along the adversary generation trajectory to achieve cross-VLM attacks."
    - This seems to be a wrong citation? Set-level Guidance Attack (SGA) does not use generation trajectory.

**Other Strengths And Weaknesses:**

Weakness:
- The presentation quality is not optimial.
  - The caption of Table 6 overlaps with the table.
  - I don’t think it’s good practice to include extensive mathematical formulations in the abstract. At the very least, using multiple notations without explanation is problematic.

**Questions For Authors:**

- Is the idea of using adversarial trajectory novel? If yes, do you think it can be applied to image classification task, not only VLMs?
- Why does the clean accuracy improve with the proposed method?
- Is the training time per epoch longer than FARE or TeCoA?
- This method uses a much larger batch size of 512 compared to 128 of TeCoA and FARE.
    - Does this method require a large batch size?
    - Do TeCoA/FARE results improve with larger batch sizes?

**Relation To Broader Scientific Literature:**

The idea of leveraging adversarial trajectory in adversarial training may be new. It could also be applied to image classification or other tasks.

**Theoretical Claims:**

- I don’t get how Eq.7 is derived. How did $\gamma(x, \delta_x)$ disappear? Are the notations correct? If $\alpha(x,\delta_x) = || J_g(x) \delta_x ||^2_2$, why is it doubled in Eq.(7)??? Sorry if I misunderstand anything.

---

> ### Author Rebuttal · Authors · 2025-03-31
>
> # Response to Rev. XcU1
> Thank you for the constructive feedback. **Kindly also note additional Theory/Results in Resp. to Rev. EyNB**.
> ### **1. Compare Cost. 10x speedup.**
> - Below are **training times for FARE, TeCoA, PMG-FT**.
> - The *10x speed-up* is for our closed-form "infinite" sampling from adv. simplex *vs.* "naive" sampling (see below *Sampling 300* samples per simplex *vs.* AdvSimplex (10 steps)).
>
> |Method|Clean|AA|Training Time per Epoch (hours)|
> |-|-|-|-|
> |TeCoA|48.83|25.75|1.3|
> |PMG-FT|49.71|26.98|2.1|
> |FARE|56.68|29.30|1.7|
> |Sampling (300 samp.)|60.18|34.20|36.2|
> |**AdvSimplex (10 steps)**|60.23|34.06|2.9|
> |**AdvSimplex (5 steps)**|58.75|33.27|1.6|
> |**AdvSimplex (3 steps)**|58.36|32.19|1.2|
> |**AdvTetrahedron (10 steps)**|60.80|35.17|2.7|
> |**AdvTetrahedron (5 steps)**|59.12|33.64|1.4|
> |**AdvTetrahedron (3 steps)**|58.67|32.70|1.1|
>
> - For **the same time budget (~1.6h), AdvSimplex (5 steps) outperforms FARE by ~2\% and ~4\%** in the clean and robust accuracy (AA)
> - For 1.7x time budget, **AdvSimplex (10 steps) gets ~3.5\% and ~4.7\% gains over FARE** (clean and robust acc. (AA)).
>
> - **Our method does not use additional modules/parameters:** the inference time is consistent with other VLMs.
>
> - **AdvTetrahedron** uses a generalization of our Theorem 3.1 to simplices with more vertices, e.g.,  **tetrahedron** ($Q=4$ vertices). **Kindly see Resp. 1 to Rev. EyNB for detailed information.**
>
> ### **2. Why Weak Adversaries can Robustify? 2b. Clean accuracy gains. 2c. SGA does not use intermediate adv. 2d. Novelty using adv. trajectory.**
>
> > [a] Set-level Guidance Attack..., Lu et al. (2023), ICCV
>
> > [b] Boosting Transferability in Vision-Language Attacks..., Gao et al.(2024), ECCV
>
> - SGA [a] in every ascent step generates few perturbations around intermediate ${\bf v}'$ and continues ascent step from their mean ($\bf\mu\neq v'$). Thus, for the same ${\bf x}$ generated 3 traj. will differ.
> - [b] forms triangles between ${\bf x}$ and consecutive 2 intermediate steps ${\bf v}\_i$ and ${\bf v}\_{i+1}$ but does not use noise. But Fig. 1b [b] shows triangles capture multiple trajectory routes.
>
> As gradient ascent uses fixed gradient step and the decision boundary is non-linear, **the adversarial path is not a perfect ascent**. Intermediate adv. samples:
> - **may be sometimes stronger than final adv. samples**
> - **enjoy diversity in adversarial directions.**
>   \
>   \
>   We verify this:  **weights $\omega\_i$ in Eq. 13 (main paper) for the final-step adversary $m$ is not always more adversarial than intermediate adversary $m-1$**:
>
>   |$\omega\_m\geq\omega\_{m-1}$|$\omega\_{m-1}\geq\omega\_{m-2}$|$\omega\_{m-2}\geq\omega\_{m-3}$|
>   |-|-|-|
>   |83%|79%|73%|
>
> *Friendly Adversarial Training (ICML'20)* notes that weak adv. help obtain robust decision boundary.
>
> Below we also show that simplices from:
> - late intermediate steps lead to greater adversarial robustness
> - early intermediate steps lead to greater clean accuracy
> - **mid intermediate steps further boost clean+adversarial accuracy**
>
> |Indices of Used Adv. Simplices|Clean|AA|
> |-|-|-|
> |8-10 (Late Steps)|58.65|33.06|
> |6-10 (Late Steps)|59.14|33.38|
> |1-10 (All)|**60.23**|**34.06**|
> |1-5 (Early Steps)|60.37|32.49|
> |1-3 (Early Steps)|59.79|32.00|
>
> **2b.** Early adversaries improve clean acc. as they are more correlated with clean ${\bf x}$ and act as sample augmentation.
>
> **2c.** To validate Rev. assumption that only final adversaries matter, per ${\bf x}$ we generate 3 adv. paths as SGA (paths differ due to noise injection). **We build a simplex only from final adv. points of 3 paths (AdvSimplex-SGA)** per ${\bf x}$: this outperforms vanilla SGA but is worse than **AdvSimplex**.
>
> |Method|Avg. Clean|Avg. AA|
> |-|-|-|
> |SGA (standard: one adv. per ${\bf x}$)|56.84|30.08|
> |SGA (3 adversaries per ${\bf x}$)|57.31|30.93 |
> |AdvSimplex-SGA (simplex on 3 adv. per ${\bf x}$)|58.75|32.40|
> |**AdvSimplex**|60.23|34.06|
>
> **2d.** Our main contribution is **"infinite" sampling+alignment but adv. simplex can be generated in many ways** (our strategy, SGA-like strategy, etc.) Unlike SGA [a] and [b] we are the first to use adv. simplices from trajectory for robustification.
>
> ### **3. $\gamma$ missing in Eq. 7**
>
> **See Resp. 1b to Rev. EyNB.**
>
> ### **4. Results differ from other papers.**
>
> - [Schlarmann, 2024] reports on a sampled subset of ImageNet (5000 samples) on **ViT-L** in their Table 4
> - We report on **ViT-B** in our Tables 1+2 following TeCoA and PMG settings (Mao, 2023; Wang, 2024)
>
>   **To address Rev.'s concern, we extend our Table 3 to ViT-L in Resp. 6 to Rev. srgi.**
>
> ### **5. Compare Batch Sizes.**
>
> In paper, we use same experimental setup (BS: 512) for TeCoA, PMG, FARE and ours for fairnesss.
> See **Resp. 7 to Rev. srgi**
>
> ### **6. Auto-Attack for Each Dataset.**
> See **table in Resp. 8 to Rev. srgi.** We can post ViT-L AA in discussions.
>
> ### **7. Extend to image classification.**
> See **Resp. 6 to Rev. td6r**
>
> ### **8. Formatting/abstract.**
> See **Resp 1-4 to  Rev. srgi**

---

> > ### Comment · Reviewer_XcU1 · 2025-04-09
> >
> > Sincere apologies, I mistakenly posted my response as an Official Comment instead of a Rebuttal Comment, which made my reply invisible to the authors.
> >
> > ---
> >
> > Thank you so much for your thorough reply. Many of the concerns are addressed.
> >
> > > 2. Why Weak Adversaries can Robustify? 2b. Clean accuracy gains. 2c. SGA does not use intermediate adv. 2d. Novelty using adv. trajectory.
> >
> > I believe this part should be incorporated into the paper. The current version starts by discussing efficient trajectory sampling but lacks an explanation of why trajectory is important, relying solely on citations. Including this discussion would enhance the paper’s quality and readability.
> >
> > Additionally, I agree with Reviewer srgi that the paper is sometimes too dense with mathematical content. I suggest revising it to provide sufficient explanations, including intuitive ones, to help readers unfamiliar with this field grasp the key ideas.
> >
> > ---
> > I will update the score.

---

> > > ### Author Response · Authors · 2025-04-09
> > >
> > > Esteemed Reviewer,
> > > \
> > > \
> > > Thank you for your kind message, and valuable comments helping us improve and refine our manuscript.
> > > \
> > > \
> > > Rest assured, all requested improvements will be made in the final paper. We will include details of Q2 as per your request, and explanations why the trajectory is important.
> > > \
> > > \
> > > Rest assured, we will revise maths accordingly to make it more accessible and intuitive and less dense.
> > > \
> > > \
> > > Kind regards,
> > > \
> > > Authors

---

### Official Review · Reviewer_td6r · 2025-03-14

**Overall Recommendation:** 4

**Summary:**

This paper proposes a new adversarial fine-tuning method for VLMs that enhances zero-shot adversarial robustness by leveraging adversarial simplices formed from a clean image and consecutive intermediate adversaries along the gradient ascent path. The proposed method employs a Taylor expansion of the model's prediction function to derive a closed-form upper bound on the alignment loss. This closed-form solution, which approximates infinite uniform sampling within the simplex, is then integrated into a combined loss function that balances standard classification loss with an adaptive adversarial alignment term, which achieves SOTA zero-shot adversarial robustness and computational efficiency across 15 datasets.

**Claims And Evidence:**

Yes, the claims made in the submission are well-supported by clear and convincing evidence.

**Essential References Not Discussed:**

None.

**Experimental Designs Or Analyses:**

The experimental design is very sound from my perspective: this paper uses 15 datasets and multiple attack methods for evaluations (e.g., PGD, C&W and AutoAttack) and all evaluations use adaptive attacks. In addition, this paper systematically measures both zero-shot clean and robust accuracy across different model architectures and perturbation budgets. The authors also provide ablation studies and sensitivity analyses that help support the validity of their choices of hyperparameters. Regarding the section on baseline methods, I notice that a recently published paper [1] can be considered as one of the baseline methods in this paper as well since it also claims that it achieves the SOTA performance in zero-shot robustness.

[1] Text-guided attention is all you need for zero-shot robustness in vision-language models, NeurIPS 2024.

**Methods And Evaluation Criteria:**

This paper aims to improve the zero-shot adversarial robustness of VLMs. Clean and robust accuracy used in this paper are widely acknowledged as the golden metrics in zero-shot robust classification problems. 15 benchmark datasets used in this paper are widely acknowledged to evaluate the zero-shot capabilities of CLIP. All the baseline methods are fine-tuned on ImageNet, which is a relatively large-scale dataset in this field.

**Other Comments Or Suggestions:**

I noticed that Table 5 and Table 6 overlap in the current layout. I would suggest adjusting their formatting to avoid overlapping.

**Other Strengths And Weaknesses:**

Strengths:

1.This paper proposes an interesting fine-tuning method that could largely improve the zero-shot performance and, at the same time, significantly reduce computational cost.

2.Theoretical derivations are well-grounded and the experimental results support the claims.

Weaknesses:

1.The experiment lacks a sensitivity analysis of the parameter $\lambda$ used in Eq. (13).

**Questions For Authors:**

1.The adaptive re-weighting mechanism in Eq. (13) appears crucial to your loss formulation. How sensitive is the method’s performance to the choice of $\lambda$?

2.In your comparisons, the closed-form alignment is shown to be computationally more efficient than explicit sampling. Could you provide further quantitative analysis on the trade-off between computational overhead and adversarial robustness when using closed-form versus sampled adversarial simplices?

**Relation To Broader Scientific Literature:**

The paper builds upon a substantial body of work in adversarial training and fine-tuning. For example, Wang et al. (2022) and Lu et al. (2023) emphasize the importance of intermediate adversarial samples along the gradient ascent path, as well as Gao et al. (2024) who introduced the concept of forming simplices with consecutive adversaries. However, previous approaches struggled with the computational costs of sampling such simplices. This paper advances the previous methods by using Taylor expansion and deriving a closed-form solution that captures the infinite sampling behavior within a simplex. This approach not only aligns with established ideas in robust optimization but also addresses practical scalability concerns, thereby contributing a novel and computationally efficient method to the field.

**Theoretical Claims:**

The proof is overall correct. In particular, I checked the derivation of Theorem 3.1 on the closed-form second-order statistics for points uniformly sampled in a simplex, as well as the bounding argument in Section 3.1 (where the paper moves from the exact alignment loss to its Taylor-based upper bound). The closed-form expression for the uniform simplex statistic (Theorem 3.1) is standard and appears correctly applied.

---

> ### Author Rebuttal · Authors · 2025-03-31
>
> # Response to Rev. td6r
> Thank you for the constructive feedback. **Kindly also note additional Theory/Results in Resp. to Rev. EyNB**.
> ### **1. Theoretical derivations well-grounded/experiments support claims.**
> Thank you. We have also a generalization of our Theorem 3.1 to simplices with more vertices, e.g.,  **tetrahedron ($Q=4$ vertices)** or **pentachoron ($Q=5$ vertices)**.
> \
> \
> **Kindly see Resp. 1 to Rev. EyNB for details.** Below we provide a list of our extensions.
>
>   Let $Q$ be the number of vertices (${\bf z}_1,\ldots,{\bf z}_Q$) for a simplex. The closed-form expression for ${\bf\Sigma}_x$ is given as:
>
>   $\mathbb{E}[{\bf pp}^T]=\frac{1}{Q(Q+1)}\bigg[\sum\_{i=1}^Q{\bf z}\_i {\bf z}\_i^T + \Bigl(\sum_{i=1}^Q {\bf z}\_i\Bigr)\Bigl(\sum_{i=1}^Q {\bf z}\_i\Bigr)^T\bigg]$.
>
> Table below shows further gains from our extended theorem:
>
> |Method|Avg. Clean|Avg. AA|
> |-|-|-|
> |Our **AdvSimplex**|60.23|34.06|
> |Our **AdvTetrahedron**|**60.80**|**35.17**|
>
> ### **2. Comparison with [1]**
> > [1] Text-guided attention is all you need for zero-shot robustness in vision-language models, NeurIPS 2024.
>
> TGA-ZSR [1] designs text-guided attention to enhance zero-shot robustness on VLMs. We will cite [1].
> \
> \
> Below we evaluate/compare our AdvSimplex with [1] on image-level, text-level, and bi-level adversarial attacks. The average results across 15 datasets are below:
>
> |||*Image-Level Attacks*||*Text-Level Attacks*||*Bi-Level Attacks*||
> |-|-|-|-|-|-|-|-|
> |**Method**|**Clean**|**PGD**|**AA**|**BERT-Attack**|**GBDA**|**Co-Attack**|**SGA**|
> |TeCoA|48.83|27.33|25.75|37.14|35.30|26.73|25.94|
> |PMG-FT|49.71|28.44|26.98|37.61|36.46|28.11|27.85|
> |FARE|56.68|30.94|29.30|35.45|34.97|25.38|25.06|
> |$\color{red}\text{TGA-ZSR [1]}$|57.54|31.15|30.41|38.07|37.32|29.30|28.58|
> |**AdvSimplex**|**60.23**|**35.68**|**34.06**|**40.21**|**39.88**|**32.95**|**32.53**|
>
> **AdvSimplex significantly outperforms other methods.**
>
> ### **3. Sensitivity w.r.t. $\lambda$.**
> - $\lambda$ controls the impact of the upper bound of $\Omega({\bf x})$ when aligning adversarial predictions with clean predictions.
> - Figure 4a (main paper) shows that increasing $\lambda$ enhances the adversarial robustness at the cost clean accuracy. Lowering $\lambda$ improves zero-shot clean acc. but reduces adv. robustness.
> - Such a trade-off stems from the optimization the natural and boundary risks.
>
> Average results across 15 datasets w.r.t. $\lambda$ are below:
>  - **Uniform weighting**: $\omega_i=1/(m-1)$ is const. for all $i=1,\ldots,m-1$.
>  - **Adaptive weighting**: as per Eq. 13 (main paper)
>
> |Re-weighting Strategy|$\quad\lambda\quad$|Clean Accuracy|AA-Robust Accuracy|
> |-|-|-|-|
> |Uniform|0.5|59.67|32.38|
> |Uniform|**0.6**|**59.45**|**32.96**|
> |Uniform|0.7|59.06|33.64|
> |Uniform|0.8|58.72|34.09|
> |Uniform|0.9|58.29|34.26|
> |Adaptive|0.5|60.41|33.35|
> |Adaptive|**0.6**|**60.23**|**34.06**|
> |Adaptive|0.7|60.02|34.75|
> |Adaptive|0.8|59.78|35.10|
> |Adaptive|0.9|59.04|35.39|
>
> In our paper, $\lambda=0.6$ in all experiments.
>
> ### **4. Formatting issue.**
> We will fix this. It appears tables jumped between pages unintentionally.
>
> ### **5. Trade-off analyses between closed-form vs. sampling.**
> - In addition to Figure 4b, we present further results (clean accuracy, adv. robust accuracy (AA-Robust Accuracy), training time per epoch):
>
> |Method|No. samples per simplex|Clean Accuracy|AA-Robust Accuracy|Train Time per Epoch (hours)|
> |-|-|-|-|-|
> |Sampling|3|57.42|28.86|1.4|
> |Sampling|10|57.63|30.37|2.6|
> |Sampling|30|57.98|31.98|5.1|
> |Sampling|50|58.55|33.17|8.0|
> |Sampling|70|59.07|33.69|10.3|
> |Sampling|100|59.61|33.92|14.0|
> |Sampling|200|59.87|34.07|25.5|
> |Sampling|300|60.18|34.20|36.2|
> |Our **closed-form simplex (10 steps)**|$\infty$|60.23|34.06|2.9|
> |Our **closed-form simplex (5 steps)**|$\infty$|58.75|33.27|1.6|
> |Our **closed-form simplex (3 steps)**|$\infty$|58.36|32.19|1.2|
> |Our **closed-form tetrahedron (10 steps)**|$\infty$|60.80|35.17|2.7|
> |Our **closed-form tetrahedron (5 steps)**|$\infty$|59.12|33.64|1.4|
> |Our **closed-form tetrahedron (3 steps)**|$\infty$|58.67|32.70|1.1|
>
> - Increasing the number of sampling adversaries from simplex yields gains in robustness at the substantial computational cost increase.
> - Our closed-form solution effectively balances robustness and computational efficiency, providing **competitive clean and robust performance at ~$10\times$ reduced computational cost.**
>
> ### **6. Extension to image classification.**
> In our paper, we also have:
> - **Vision-text understanding.** Table 7 is **image-text retrieval (Flickr30k)** and **image captioning (Nocaps)**
> - **Medical diagnosis.** Table 8 shows results on ChestX-ray14, CheXpert and PadChest.
>
> For more results, table below is single-modal image classification built on TRADES with Wide-ResNet-28-10 (CIFAR-10/100):
> ||CIFAR-10||CIFAR-100||
> |-|-|-|-|-|
> |**Method**|**Clean**|**AA**|**Clean**|**AA**|
> |PGD-AT|83.79|56.30|58.13|26.65|
> |TRADES|85.17|57.09|59.48|26.99|
> |**AdvSimplex**|**87.25**|**58.64**|**61.30**|**28.57**|

---

> > ### Comment · Reviewer_td6r · 2025-04-03
> >
> > I would like to thank the authors for their comprehensive rebuttal. My concerns are **well-addressed** and I will keep my score towards accepting this paper. Please add these additional experiments to the paper, which I believe will make the paper more solid. Best of luck with the rebuttal!

---

> > > ### Author Response · Authors · 2025-04-03
> > >
> > > Esteemed Reviewer,
> > > \
> > > \
> > > Thank you for your kind message, and valuable comments helping us improve and refine our manuscript. Meantime, if there is anything else we can answer or explain or discuss further, kindly do let us know.
> > > \
> > > \
> > > Rest assured, all requested explanations and details will be added in the final paper.
> > >
> > > Kind regards,
> > > \
> > > Authors

---

### Official Review · Reviewer_EyNB · 2025-03-20

**Overall Recommendation:** 5

**Summary:**

This paper tackles the challenge of making vision-language models (VLMs) more robust to adversarial attacks in zero-shot classification. Existing methods try to improve robustness by sampling adversarial examples along the decision boundary, but they come with high computational costs. To address this, the authors propose a more efficient approach using a closed-form formulation based on Taylor expansion. This allows them to approximate adversarial alignment loss without expensive second-order computations while still capturing key adversarial perturbations along the trajectory. Their method effectively simulates infinite uniform sampling over simplices, offering a computationally efficient way to strengthen VLMs against attacks.

**Claims And Evidence:**

All claims in the paper are clearly stated and supported by theoretical proofs.

**Essential References Not Discussed:**

The paper covers the essential references relevant to its topic, and I did not find any major prior work that was overlooked.

**Experimental Designs Or Analyses:**

Yes, I reviewed the experimental design and analyses, which appear well-structured and aligned with the paper’s objectives. The evaluation spans 15 datasets, providing a diverse testbed for assessing the robustness of the proposed method. The comparisons with prior adversarial fine-tuning approaches are relevant, and the reported 10× speed-up adds an important efficiency perspective.

**Methods And Evaluation Criteria:**

The proposed method is well-motivated and effectively addresses the computational challenges of adversarial fine-tuning for vision-language models (VLMs). The use of a closed-form formulation based on Taylor expansion makes sense for improving efficiency while maintaining robustness. The evaluation criteria are appropriate, with experiments conducted diverse tasks and datasets, which provides a strong empirical results for assessing the method’s effectiveness.

**Other Comments Or Suggestions:**

NA

**Other Strengths And Weaknesses:**

NA

**Questions For Authors:**

NA

**Relation To Broader Scientific Literature:**

This paper builds on prior work in adversarial robustness for vision-language models (VLMs) and extends research on efficient adversarial training methods. Traditional adversarial fine-tuning methods, such as those explored by Wang et al. and Lu et al., have highlighted the importance of intermediate adversarial examples along the adversarial trajectory for improving robustness. However, these approaches rely on costly second-order computations, limiting their scalability. The proposed method addresses this limitation by leveraging Taylor expansion to approximate adversarial alignment loss in a closed-form solution, eliminating the need for expensive sampling of adversarial simplices.

**Theoretical Claims:**

Regarding the theoretical claims, the paper provides clear mathematical derivations to support its approach. The Taylor expansion and the closed-form approximation of adversarial alignment loss are well-structured and logically follow from the given assumptions. I checked the key proofs related to the approximation and alignment loss formulation, and they appear to be correct.

---

> ### Author Rebuttal · Authors · 2025-03-31
>
> # Response to Rev. EyNB
> We thank Rev. for constructive feedback.
> ### **1. I checked key proofs...they appear to be correct.**
>
> Thank you.
>
> ### **1b.** Why $\gamma$ disappears in Eq. 7? (for Rev. XcU1)
>
> - In Eq. 5 \& 6, elements after the sum can be written as $(\sqrt{\alpha}+\frac{1}{2}\sqrt{\beta})^2=\alpha+\frac{1}{4}\beta+2\cdot\frac{1}{2}\gamma$ where $\gamma=\sqrt{\alpha\beta}\;,\alpha,\beta\geq 0$.
> - Use known inequality $(a+b)^p \le 2^{p-1}(a^p+b^p)$ and set $p=2$ so $(a+b)^2 \le 2^{1}(a^2+b^2)$.
> - Substitute the first eq. into the inequality: $(\sqrt{\alpha}+\frac{1}{2}\sqrt{\beta})^2\le 2\alpha +2\cdot\frac{1}{4}\beta=2\alpha +\frac{1}{2}\beta$ and that is why $\gamma$ does not appear in Eq. 7.
>  - This result corresponds to $\bar{\bar{\Omega}}$ in Eq. 10 which is an upper bound of Eq. 6.
>  - Note we evaluate in supp. material (**C.2**) Eq. 9 includes $\gamma$ but is very costly due to operations in $\mathbb{R}^{(wh)^2}$ *vs.* $\mathbb{R}^{wh}$.
>
> ### **1c. To further showcase our work, below we have additional extensions of Theorem 3.1.**
>
> - **Theorem 3.1 (extended)** can be extended to higher-order simplices, e.g., **tetrahedron ($Q=4$ vertices)** or **pentachoron ($Q=5$ vertices)**:
>
>   Let $Q$ be the number of vertices (${\bf z}_1,\ldots,{\bf z}_Q$) for a simplex. The closed-form expression for ${\bf\Sigma}_x$ is given as:
>
>   $\mathbb{E}[{\bf pp}^T]=\frac{1}{Q(Q+1)}\bigg[\sum\_{i=1}^Q{\bf z}\_i {\bf z}\_i^T + \Bigl(\sum_{i=1}^Q {\bf z}\_i\Bigr)\Bigl(\sum_{i=1}^Q {\bf z}\_i\Bigr)^T\bigg]$.
>
>   Proof follows steps in paper, e.g., parameterizing ${\bf p}=\sum\_i^Q\alpha_n{\bf z}\_i, \\;\alpha\_i\geq 0,\\; \sum\_i^Q\alpha\_i=1$ and noting that:
>   - $\mathbb{E}[\alpha_i] = \frac{1}{Q}$
>   - $\mathbb{E}[\alpha_i^2] = \frac{2}{Q(Q+1)}$
>   - $\mathbb{E}[\alpha_i\alpha_j] = \frac{1}{Q(Q+1)},\\;i\neq j$
>
>   for the underlying Dirichlet distribution. Then we expand $\mathbb{E}[pp^T]$ where $p p^T = \sum\_{i,j=1}^Q \alpha\_i\alpha\_j {\bf z}\_i{\bf z}\_j^T$.
>
> ### **1d** We have also a theoretical extension linking Theorem 3.1 above with the **Hessian-vector product (HVP)** needed for fast evaluation of Eq. 10.
>
> Evaluating Hessian even with **functorch** is slow. In our paper, **we used HPV which never evaluates Hessian explicitly.** HPV computes very fast $({\bf H\_g\cdot v})$ instead.
>
> - To take advantage of it, Theorem 3.1 includes:
>
>     $\mathbb{E}[{\bf p}^T (H\_g\cdot{\bf p})]=\frac{1}{Q(Q+1)}\bigg[\sum\_{i=1}^Q{\bf z}\_i^T(H\_g\cdot{\bf z}\_i) + \Bigl(\sum_{i=1}^Q {\bf z}\_i\Bigr)^T\Bigl(H\_g\cdot\sum_{i=1}^Q {\bf z}\_i\Bigr)\bigg]$.
>
> - Thus, in Eq. 10, we can substitute $\langle {\bf \Sigma}\_x, (H\_g({\bf x}))\_c\rangle^2=\Big(\mathbb{E}[{\bf p}^T ((H\_g({\bf x}))\_c\cdot{\bf p})]\Big)^2$. For a simplex with three vertices (one is $0$), this requires three HPV evaluations.
>
> ### **2. Alternative to Euclidean alignment in Eq. 1.**
>
> We replace $\Omega({\bf x})$ in Eq. 1 with a KL-divergence variant integrated with Theorem 3.1. To this end:
> - we use a different Taylor expansion $\log g({\bf x}+{\bf\delta}\_x)-\log g({\bf x})\approx J\_{\log (g+\rho)}({\bf x}){\bf\delta}\_x + \frac{1}{2}\bigl[{\bf\delta}\_x^T(H\_{\log (g+\rho)}({\bf x}))\_c\\,{\bf\delta}\_x\bigr]_{c=1}^{C}$ where $\rho=1e^{-5}$ is added for the numberical stability of $\log$.
> - then $\Omega\_{KL}({\bf x})=\frac{1}{\kappa}\sum\nolimits\_{{\bf\delta}\_x\in\Delta\_\mathcal{X}} KL\big(g({\bf x}) || g({\bf x}+{\bf\delta}\_x)\big)\approx-\sum\_{c=1}^C (g({\bf x}))\_c\cdot \Big[  \big\langle{\bf\mu}\_x, \mathcal{J}\_{\log (g+\rho)}({\bf x},c) \big\rangle+\frac{1}{2}\big\langle{\bf{\Sigma}}\_x, (H\_{\log (g+\rho)}({\bf x}))\_c \big\rangle \Big]$
>
>   where ${\bf\mu}\_x$ is the analytical mean of simplex, and ${\bf{\Sigma}}\_x$ is from Theorem 3.1.
>
> ### **3. Results for extensions.**
>
> |Method|Avg. Clean|Avg. AA|Train Time per Epoch (hours)|
> |-|-|-|-|
> |KL Div. in Eq. 1 (sampling)|59.28|32.96|3.5|
> |AdvSimplex-KL|59.13|33.20|3.0|
> |SGA (standard: one adv. per ${\bf x}$)|56.84|30.08|1.3|
> |SGA (3 adversaries per ${\bf x}$)|57.31|30.93 |1.4|
> |AdvSimplex-SGA (simplex on 3 adv. per ${\bf x}$)|58.75|32.40|2.1|
> |**AdvSimplex**|60.23|34.06|2.9|
> |**AdvTetrahedron**|60.80|35.17|2.7|
>
> In the table above:
> - **AdvSimplex-KL** uses our KL div. $\Omega\_{KL}({\bf x})$ instead of bounds of Euclidean $\Omega({\bf x})$.
> - **AdvTetrahedron** uses our extended Theory 3.1 for tetrahedron ($Q=4$). We take ${\bf x}$ and additional consecutive 3 vertices instead of 2 as in AdvSimplex ($Q=3$).
> - **AdvSimplex-SGA** uses simplex formed from the final 3 adversaries of 3 adversarial paths per sample ${\bf x}$. As SGA perturbs each intermediate adversary by noise, running gradient ascent 3x produces 3 distinct paths.
> - **SGA (3 adv.)**: we generate 3 adv. paths as in AdvSimplex-SGA and use the final 3 adversaries directly for robustification.
> - **SGA (standard)**: we use the final adversary from a single adv. path for robustification.
>
> SGA:
> > Set-level Guidance Attack..., ICCV'23.

---

### Decision · Program_Chairs · 2025-05-01

**Decision:**

Accept (spotlight poster)

**Comment:**

This paper proposes an innovative closed-form alignment of adversarial path simplices to further improve the zero-shot classification performance of a pre-trained VLM. The main contributions of this paper come from two sides: 1) it is novel to consider using path simplices for the adversarial robustness and 2) solid theoretical justifications make this paper effective in more scenarios.

In the initial reviews, two reviewers find some confusing parts within the paper (experiments and presentation). Then, the authors did a great job in addressing these concerns, and all reviewers agreed to accept this paper in the end.